# Boost Self-Supervised Dataset Distillation via Parameterization, Predefined Augmentation, and Approximation

**Sheng-Feng Yu**[12]**, Jia-Jiun Yao**[1]**, Wei-Chen Chiu**[1]

National Yang Ming Chiao Tung University[1], Macronix International Co., Ltd.[2]
`robertyu1@mxic.com.tw`, {`tony.yao.10, walon`}`@nycu.edu.tw`

## Abstract

Although larger datasets are crucial for training large deep models, the rapid growth of dataset size has brought a significant challenge in terms of considerable training costs, which even results in prohibitive computational expenses. Dataset Distillation becomes a popular technique recently to reduce the dataset size via learning a highly compact set of representative exemplars, where the model trained with these exemplars ideally should have comparable performance with respect to the one trained with the full dataset. While most of existing works upon dataset distillation focus on supervised datasets, we instead aim to distill images and their self-supervisedly trained representations into a distilled set. This procedure, named as Self-Supervised Dataset Distillation, effectively extracts rich information from real datasets, yielding the distilled sets with enhanced cross-architecture generalizability. Particularly, in order to preserve the key characteristics of original dataset more faithfully and compactly, several novel techniques are proposed: 1) we introduce an innovative parameterization upon images and representations via distinct low-dimensional bases, where the base selection for parameterization is experimentally shown to play a crucial role; 2) we tackle the instability induced by the randomness of data augmentation – a key component in self-supervised learning but being underestimated in the prior work of self-supervised dataset distillation – by utilizing predetermined augmentations; 3) we further leverage a lightweight network to model the connections among the representations of augmented views from the same image, leading to more compact pairs of distillation. Extensive experiments conducted on various datasets validate the superiority of our approach in terms of distillation efficiency, cross-architecture generalization, and transfer learning performance.

## 1 Introduction

In the realm of deep learning, the inexhaustible need for extensive datasets, e.g. ImageNet (Deng et al., 2009) and LAION (Schuhmann et al., 2022), for model training is a double-edged sword. On one hand, large datasets are generally instrumental in training better-performing models; on the other hand, they introduce prohibitive training costs. Moreover, some learning scenarios even require repeated or iterative training processes, such as continual learning and neural architecture search, thus further inflating the expense of training upon large datasets and leading to the financial burden of computing power. Hence, as the volume of data required for deep learning increases, so does the need to innovate the ways to curtail these expanding training expenses.

To this end, *Dataset Distillation* (DD) (Wang et al., 2018; Zhao & Bilen, 2021b; Cazenavette et al., 2022; Kim et al., 2022) and *coreset selection* (Coleman et al., 2020) emerge as two promising strategies for dataset reduction. Coreset selection focuses on identifying the most representative samples within a dataset. These samples, named coresets, contribute more significantly to performance than the others, allowing models trained on coresets to achieve higher performance compared to those trained on a random subset of the same size. Conversely, DD aims to create a distilled dataset, which is synthetic and optimized to maintain the model performance (i.e. the model trained upon distilled data should perform comparably to the one resulted from the full dataset). Despite the synthetic

nature of the distilled data, which results in a distribution that differs from the original data, DD has been shown to substantially improve training outcomes compared to coreset selection, albeit at the cost of longer processing times.

The concept of DD was introduced by Wang et al. (2018). It employs a bilevel optimization (composed of inner and outer loops), where the inner loop utilizes distilled data to train a model, and the outer loop optimizes the distilled data based on the model predictions on real data. Subsequent researches alter the optimization target, as seen in methods like DM (Zhao & Bilen, 2023) and MTT (Cazenavette et al., 2022). DM aims to ensure that the mean of the distilled data matches the mean of the real data, while MTT focuses on aligning the training trajectory between the real and distilled data. While earlier works on DD have achieved notable success in reducing dataset size without sacrificing model performance, most of these efforts have been limited to supervised datasets, named as supervised DD. Supervised DD, which relies heavily on labeled data, tends to emphasize class-discriminative features while neglecting others. This narrow focus can result in overfitting to specific models, limiting cross-architecture generalization and reducing effectiveness in other tasks. Consequently, this approach often struggles with transferability to downstream applications. In contrast, self-supervised learning (Grill et al., 2020; Zbontar et al., 2021; Ericsson et al., 2021), has demonstrated its ability to map images to representations, that effectively transfer to a variety of downstream tasks. Thus, condensing these generalized representations offers the potential to create a distilled dataset which superior cross-architecture generalization and improved transferability.

A recent study, KRR-ST (Lee et al., 2024), proposes the first self-supervised DD framework for transfer learning, aiming to distill a set of image and representation pairs from an unlabeled image dataset. The distilled dataset can be used to train a new model that is encouraged to mimic the self-supervised model trained on the entire unlabeled dataset, where the resultant new model could act as a good initialization for being transferred to another task through finetuning. Basically, the distillation framework in KRR-ST is as follows: 1) First, a teacher model is self-supervisedly trained on the unlabeled dataset; 2) A bilevel optimization process is then adopted, where the distilled image-representation pairs are used to train an inner model in the inner loop via minimizing the mean square error (i.e., given an input image, the representation extracted by the model should be close to its paired one), while the outer loop aims to update the distilled pairs by aligning the inner model representations with the teacher model. It is also worth noting that KRR-ST discovers that the random data augmentation (which typically is a crucial component in self-supervised learning algorithms) is incompatible with the bilevel optimization, where the gradient bias stemmed from random data augmentations would affect the optimization process, in results KRR-ST proposes to bypass the augmentation operations in its optimization to avoid such issue of incompatibility.

Although being the pioneer to tackle the self-supervised dataset distillation, KRR-ST still requires substantial storage space for the distilled images and their paired representations, which clearly are not compact enough thus constraining the overall distillation efficiency. To this end, in this work we highlight two key issues of KRR-ST and address them with our proposed techniques:

1. Each of the distilled images (respectively, their corresponding representations) in KRR-ST are stored independently, where the redundancy nor the regularity among images (respectively, representations) are not taken into consideration. As inspired by Deng & Russakovsky (2022), it has shown that learning the common bases shared among distilled images not only benefits to compress them (as images are now represented by coefficients, i.e. *parameterization*) but also contributes to improve performance, as well as the observation upon *dimensional collapse* found in Jing et al. (2022) and Hua et al. (2021) where the representation trained from self-supervised learning is distributed only in a low-dimensional subspace, we propose to separately parameterize both images and representations into two sets of bases (namely image bases and representation bases) with all distilled images and representations being reconstructed through their corresponding bases. In addition, we find that the initialization of the bases is the key to obtaining better model performance, hence we propose initializing the bases using the *principal components* of the given real dataset (i.e. the source of distillation).

2. As previously mentioned, KRR-ST skips the data augmentation during its optimization process, which is however a critical component in the realm of self-supervised learning. In turn, we propose to keep leveraging the benefits of data augmentation via predefining all

possible augmentations and storing all of the representations of augmented images to avoid the randomness (which would lead to gradient bias in the bilevel optimization). Moreover, as independently storing the representations of all augmented views of the same image consumes considerable storage space, we introduce the approximation networks (built upon multiple-layer perceptrons) which learn to predict the shifts in terms of representation from the original distilled image to its augmented views.

With our two proposed techniques as described above, we only need to save the bases of distilled images and representations along with their coefficients, as well as the approximation networks (cf. Figure 1 for an overview of our method).

We experiment our proposed method by condensing a source dataset (i.e. CIFAR100 (Krizhevsky, 2009), TinyImageNet (Le & Yang, 2015), and ImageNet (Deng et al., 2009)) into a small distilled dataset. Subsequently, to evaluate the quality of the distilled dataset, a feature extractor (whose architecture could be different from the model used in the inner loop of the distillation procedure) is firstly pretrained on this distilled dataset, followed by performing linear evaluation of such feature extractor on the source dataset or the target datasets (e.g., CIFAR10 (Krizhevsky, 2009), CUB2011 (Wah et al., 2011), Stanford Dogs (Khosla et al., 2011)). Our proposed method produces a superior distilled dataset, demonstrating improved linear evaluation results across various target datasets and feature extractor architectures. Our key contributions are summarized as follows:

- We introduce efficient parameterization for both distilled images and corresponding representations, coupled with a novel use of predefined data augmentations and approximation networks, to further improve the compactness of the distilled dataset.

- Our distilled dataset experimentally demonstrates better cross-architecture generalizability and the linear evaluation results across various image datasets.

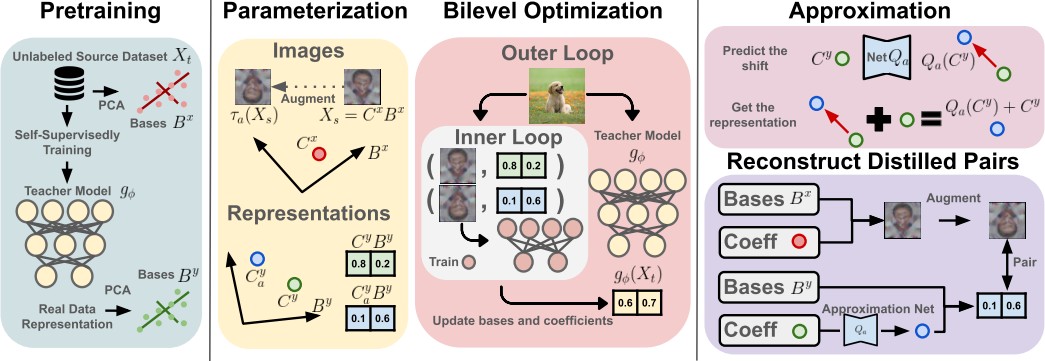

Figure 1: The illustration of our proposed framework of self-supervised dataset distillation. **Pretraining**: A teacher model $g_\phi$ is self-supervisedly trained on the unlabeled source data $X_t$ (i.e. the real dataset which we would like to distill). The initialization of the image bases $B^x$ is based on the principal components of $X_t$, and the principal components of $g_\phi(X_t)$ (i.e. the representations of real data extracted by the teacher model $g_\phi$) are adopted to initialize the representation bases $B^y$. **Parameterization**: The distilled images $X_s$ are parameterized by the linear combination of image bases $B^x$ with coefficients $C^x$, and the corresponding representations are defined by the linear combination of the representation bases $B^y$ with coefficients $C^y$. As for the augmented images $\tau_a(X_s)$, we parameterize its representation by creating a new coefficient $C_a^y$. **Bilevel Optimization**: The distilled data and its augmentations are used to train a feature extractor within the inner loop. In the outer loop, the feature extractor is encouraged to have a similar output as $g_\phi$. **Approximation**: Approximation networks $Q_a$ are trained to predict the representation shift $C_a^y - C^y$ caused by the augmentation, and we get the approximate representation coefficient of the augmented image by summing the predicted shift $Q_a(C^y)$ and $C^y$. **Reconstruct Distilled Pairs**: For the future use, we can reconstruct the distilled data from image bases, representation bases, image coefficients, representation coefficients, and the approximation networks.

## 2 RELATED WORK

**Self-Supervised Learning.**

While unsupervised learning refers to the general scheme of learning from data without human supervision, self-supervised learning (SSL), a type of unsupervised learning, uses tasks designed by the model itself to learn from the data. The learned features from SSL are usually good at generalizing and transferring to various downstream tasks (Ericsson et al., 2021). Early attempts of SSL often introduce pretext tasks of recognizing the transformation (e.g. image shuffling or rotation) applied upon the input, effectively leading the model to acquire the capability of differentiating between images. In recent years, contrastive learning stands out as a particularly promising strategy of SSL (Chen et al., 2020; Grill et al., 2020; Chen & He, 2021; Bardes et al., 2021; Zbontar et al., 2021), where the model learns knowledge through distinguishing between similar (positive) and dissimilar (negative) pairs of data samples. For example, SimCLR (Chen et al., 2020) generates two versions of an image and teaches the model to bring similar pairs closer and push different pairs apart. However, it needs many negative (dissimilar) samples to improve accuracy. To circumvent this challenge, Barlow Twins (Zbontar et al., 2021) offers a novel approach to SSL that eliminates the requirement for negative samples. It aims to reduce feature redundancy between differently augmented versions of the same image, aligning their correlation matrix with the identity matrix to ensure diverse and efficient representation learning. In this work, Barlow Twins is used as the SSL method to train the teacher model.

**Dataset Distillation.** The need of extensive training datasets for modern deep learning models has made training cost reduction a key research area. Dataset distillation (DD), initially introduced by Wang et al. (2018), seeks to create compact informative samples that can effectively train deep learning image classification models. Within a constrained storage budget, it formulates DD as a bilevel meta learning problem where the outer loop optimizes the distilled images by minimizing the classification loss on real training data, and the inner loop trains a model on the distilled data. Subsequent studies have expanded on this concept by introducing I) various matching criteria, II) simplifying bilevel optimization, and III) image parameterization under a limited storage budget.

I) Matching criteria can be broadly classified into two categories. The first approach focuses on changes in the model's parameters when trained on real data versus distilled data. Gradient matching techniques (Zhao & Bilen, 2021b;a) aim at mimicking the one-step gradient direction of actual training images, but tend to falter in accuracy due to their short-term focus. In contrast, MTT (Cazenavette et al., 2022) and DATM (Guo et al., 2024), trajectory matching methods, enhances performance by updating the distilled images to minimize the difference in terms of networks' parameters (i.e. the network trained on distilled images versus the one trained on real data) over several training epochs, addressing the limitations of gradient matching by considering the longer-term training dynamics. The second category shifts focus towards the characteristics of data itself. These methods aim to ensure that the distilled images reflect the distribution of real training samples, with techniques like DM (Zhao & Bilen, 2023) and IDM (Zhao et al., 2023) focusing on aligning the means of each class.

II) Another recent advancements have focused on streamlining the complex bilevel optimization associated with DD. Approaches based on kernel methods (Nguyen et al., 2021; Zhou et al., 2022; Loo et al., 2022; 2023) aim to derive a closed-form solution for optimization within the inner loop. Notably, FRePo (Zhou et al., 2022) concentrates on training only the final layer of a neural network to its convergence, maintaining a static feature extractor. This methodology allows inner optimization to be efficiently represented as Kernel Ridge Regression (KRR) (Murphy, 2012), significantly reducing computational demands.

III) Image parameterization (Kim et al., 2022; Deng & Russakovsky, 2022; Liu et al., 2022; Wei et al., 2023; Shin et al., 2023) seeks for a low-dimensional space, aiming not only to compress the size of distilled images but also to narrow the optimization search space for these images. RTP (Deng & Russakovsky, 2022) explores image factorization into shared bases, then recall these bases to synthesize the distilled image, thereby minimizing redundant information across different classes and enabling a more efficient utilization of constrained storage capacities.

Beyond the supervised setting, DD in the self-supervised setting is under exploration. A recent publication, KRR-ST (Lee et al., 2024), introduces the pioneering framework for self-supervised

DD tailored for transfer learning. This framework aims to distill a set of image and representation pairs from an unlabeled dataset, and the distilled pairs have the ability to train a new model that has a similar representation ability as what is trained on the full unlabeled dataset. Moreover, they prove that the bilevel optimization procedure widely used in DD is incompatible with the random data augmentations, which play a critical role in self-supervised learning. In detail, a gradient of synthetic samples with respect to a SSL objective in naive bilevel optimization is biased due to the randomness originating from data augmentations or masking. To eliminate gradient bias, the data augmentation technique cannot be involved in the bilevel optimization procedure. Hence, in the inner loop they train a model (i.e. inner model) by minimizing the mean squared error (MSE) between predicted representations of the synthetic examples and their corresponding target feature representations without using any augmentation. To ensure that the model attains a similar representational capacity to the teacher model which self-supervisedly pretrains on the full unlabeled dataset, they also introduce the MSE between representations of the inner model and the teacher model on the original full dataset for the optimization of outer loop. Finally, when other layers are fixed, optimization is restricted to the last layer of the inner model, which can be efficiently resolved by KRR (Murphy, 2012), offering a closed-form solution for the linear head. In this work, we focus on how to apply both image and representation parameterization in self-supervised DD, as well as releasing the constraints upon augmentations.

## 3 METHODOLOGY

### 3.1 PRELIMINARY

**Problem Definition.** The problem of *self-supervised dataset distillation* for image dataset is the process to create the synthetic dataset $X_s = [\tilde{x}_1, \ldots, \tilde{x}_m]^\top \in \mathbb{R}^{m \times d_x}$ which consists of $m$ images with size $d_x = c \times h \times w$ (for simplicity, we omit the process that reshape the image with size $c \times h \times w$ into a vector with length $d_x$, and vice versa) and $Y_s = [\tilde{y}_1, \ldots, \tilde{y}_m]^\top \in \mathbb{R}^{m \times d_y}$ which have $m$ target representations (corresponding to $X_s$) with dimension $d_y$. Such synthetic dataset preserves most of the information from the unlabeled dataset $X_t = [x_1, \ldots, x_n]^\top \in \mathbb{R}^{n \times d_x}$, while keeping $m$ is significantly less than the size of the unlabeled dataset $n$. The aim here is to expedite the pretraining phase of a neural network with any architecture by employing the distilled dataset $(X_s, Y_s)$ as a substitute for the unlabeled dataset $X_t$ to perform model training.

To achieve this, a teacher model $g_\phi : \mathbb{R}^{d_x} \to \mathbb{R}^{d_y}$ with parameters $\phi$ is self-supervised pretrained on the unlabeled dataset $X_t$, where the teacher model $g_\phi$ maps input samples to their $d_y$-dim representations. Subsequently, a new model $\hat{g}_\theta : \mathbb{R}^{d_x} \to \mathbb{R}^{d_y}$ with parameters $\theta$ is trained on the distilled dataset $(X_s, Y_s)$, and the output of $\hat{g}_\theta$ is encouraged to mimic the output of teacher model $g_\phi$ by adjusting $(X_x, Y_s)$. This process can be formulated as a bilevel optimization:

$$\min_{X_s, Y_s} \mathcal{L}_{\text{outer}}(\theta^*; X_t, \phi), \text{ where } \theta^*(X_s, Y_s) = \arg\min_\theta \mathcal{L}_{\text{inner}}(\theta; X_s, Y_s) \tag{1}$$

where $\mathcal{L}_{\text{inner}}$ is the objective used to train the model $\hat{g}_\theta$, while $\mathcal{L}_{\text{outer}}$ is the objective which encourages that the output distribution of $\hat{g}_{\theta^*}$ and the teacher model $g_\phi$ are close on all real data $X_t$. The effectiveness of the distilled dataset $(X_s, Y_s)$ is assessed by applying it to train a random initialized neural network and evaluating this network to a range of downstream tasks through linear evaluation.

**Kernel Ridge Regression on Self-Supervised Target (KRR-ST)** (Lee et al., 2024) KRR-ST represents an self-supervised dataset distillation technique that employs bilevel optimization to distill the unlabeled dataset $X_t$. In the context of the bilevel optimization framework, it is theoretically proven that random data augmentation biases the outer loss with respect to the distilled dataset $(X_s, Y_s)$. To avoid randomness, this approach suggests the exclusion of all data augmentations, leveraging the mean square error (MSE) within the bilevel optimization.

In the inner loop, the feature extractor $\hat{g}_\theta$ is trained to adapt to the distilled data $(X_s, Y_s)$ by minimising MSE between the predicted representation of distilled data $\hat{g}_\theta(X_s) = [\hat{g}_\theta(\tilde{x}_1), \ldots, \hat{g}_\theta(\tilde{x}_m)]^\top$ and its corresponding target $Y_s$:

$$\mathcal{L}_{\text{inner}}(\theta; X_s, Y_s) = \|\hat{g}_\theta(X_s) - Y_s\|_F^2 = \frac{1}{m} \sum_{i=1}^m \|\hat{g}_\theta(\tilde{x}_i) - \tilde{y}_i\|^2 \tag{2}$$

In the outer loop, the goal is to optimize the distilled data $(X_s, Y_s)$, ensuring that the representation upon real data extracted by the inner feature extractor could closely align with the one obtained by the teacher model $g_\phi$. This optimization problem is formulated as the MSE between $\hat{g}_\theta(X_t)$ and $g_\phi(X_t)$ where $\hat{g}_\theta(X_t) = [\hat{g}_\theta(x_1), \ldots, \hat{g}_\theta(x_n)]^\top \in \mathbb{R}^{n \times d_y}$ and $g_\phi(X_t) = [g_\phi(x_1), \ldots, g_\phi(x_n)]^\top \in \mathbb{R}^{n \times d_y}$. However, minimizing the MSE requires backpropagation through the entire inner loop, introducing significant computational and memory overhead. FRePo (Zhou et al., 2022) suggests that the model $\hat{g}_\theta$ can be seen as the composition of a feature extractor $f_\omega$ and a linear head $h_W$, that is $\hat{g}_\theta = h_W \circ f_\omega$, and training only the linear head $h_W$ to convergence while keeping other layers (i.e. $f_\omega$) fixed. This optimization can be efficiently solved with a closed form solution, known as kernel ridge regression (KRR) (Murphy, 2012), in which the outer loop objective becomes:

$$\mathcal{L}_{\text{outer}}(X_s, Y_s; f_\omega) = \frac{1}{2}\|g_\phi(X_t) - f_\omega(X_t)f_\omega(X_s)^\top(K_{X_s,X_s} + \lambda I_m)^{-1}Y_s\|_F^2 \tag{3}$$

where $\lambda$ is a hyperparameter for regularization, $I_m \in \mathbb{R}^{m \times m}$ is an identity matrix, $K_{X_s,X_s} = f_\omega(X_s)f_\omega(X_s)^\top \in \mathbb{R}^{m \times m}$ is the Gram matrix of all distilled samples, and $f_\omega(X_s) = [f_\omega(\tilde{x}_1), \ldots, f_\omega(\tilde{x}_m)]^\top$.

Moreover, KRR-ST employs multiple models in inner optimization, enhancing robustness against overfitting compared to the use of a single model (Cazenavette et al., 2022; Zhou et al., 2022; Zhao et al., 2023). Building on this concept, KRR-ST introduces a model pool $\mathcal{M}$, comprising $L$ feature extractors. To initialize each feature extractor $\hat{g}_\theta$ in the pool $\mathcal{M}$, they first sample a training step $z \in \{1, \ldots, Z\}$, where $Z$ represents the maximum step count. Subsequently, $\theta$ is optimized to minimize the MSE on $X_s$ and $Y_s$, using gradient descent for $z$ steps. Afterward, the trained feature extractors and the their trained steps are stored in the model pool, denoted by $\mathcal{M} = \{(\hat{g}_{\theta_1}, z_1), \ldots, (\hat{g}_{\theta_L}, z_L)\}$, and a feature extractor is random sampled from $\mathcal{M}$ at the start of each inner loop.

### 3.2 IMAGE AND REPRESENTATION PARAMETERIZATION

Dataset distillation seeks to encapsulate maximal information within a single distilled image. However, image data frequently contains redundant elements or has regularity between adjacent pixels. For decades, the pursuit within the computer vision research community has been to identify effective methods for dimensional reduction of images. A notable method, Eigenface (Turk & Pentland, 1991), utilizes *principal component analysis* (PCA) to decompose human face images into a set of basis vectors, known as eigenfaces. These eigenfaces have proven to be valuable features for facial recognition and classification tasks. Furthermore, a linear combination of eigenfaces can accurately represent a wide variety of human faces.

Inspired by this concept, our aim is to parameterize the distilled image $X_s$ into a set of bases $B^x$ and a set of coefficients $C^x$, where the latter is used to linearly combine the bases to generate distilled images. The initialization of the bases comes from the first $U$ largest principal components obtained by applying PCA upon the unlabeled images $X_t$. Noting that in order to further reduce the dimension of the bases, we scale down the image size from $d_x = c \times h \times w$ to $d_x^b = c \times \frac{h}{s} \times \frac{w}{s}$ where $s$ is the scaling factor (following the practice in Kim et al. (2022)). For the initialization of coefficients, we randomly select $m$ images from the unlabeled dataset $X_t$, denoted as $X_t' = [x_1', \ldots, x_m']^\top \subset X_t$. The initial coefficients are determined by projecting the images in $X_t'$ onto the $U$ bases.

In detail, the parameterization of $m$ distilled images $X_s \in \mathbb{R}^{m \times d_x}$ is formulated as $X_s = D(C^x B^x, s)$ where $B^x = [b_1^x, \ldots, b_U^x]^\top \in \mathbb{R}^{U \times d_x^b}$ are the image bases with $b_u^x$ representing the $u$-th largest principal component of the dataset $X_t$ for $u = 1, \ldots, U$, $C^x = [c_1^x, \ldots, c_m^x]^\top \in \mathbb{R}^{m \times U}$ are coefficients with $c_i^x$ being the weight of image $x_i'$ which is projected onto the bases $B^x$ for $i = 1, \ldots, m$, and $D(\cdot, s)$ is the upsampling function that upsamples the image from $d_x^b$ to $d_x$.

Additionally, the representation obtained via self-supervised learning methods suffers from dimensional collapse (Jing et al., 2022; Hua et al., 2021), which means that the representation ends up spanning a low-dimensional subspace instead of the entire available embedding space. Following this idea, our goal is to build up the bases $B^y$ of the feature representations for the distilled dataset, where we apply PCA upon real data representation $g_\phi(X_t)$ and take the resultant first $V$ largest principal component as the initialization for $B^y$. Regarding the initialization for corresponding coefficients $C^y$ of $Y_s$ with respect to $B^y$, we also adopt the $m$ images $X_t'$ (which are used to initialize $C^x$ as described in the previous paragraph) followed by taking the weights of projecting $g_\phi(X_t')$

onto the bases $B^y$ as the initial $C^y$. That is, $Y_s = C^y B^y$, where $C^y = [c_1^y, \ldots, c_m^y]^\top \in \mathbb{R}^{m \times V}$ are the coefficients and $B^y = [b_1^y, \ldots, b_V^y]^\top \in \mathbb{R}^{V \times d_y}$ contains $V$ bases with dimension $d_y$.

## 3.3 Predefined Data Augmentation and Feature Approximation

Although KRR-ST (Lee et al., 2024) points out that the "random" data augmentation would lead to gradient bias in bilevel optimization thus abandoning the usage of augmentation, here we advance to relief such limitation upon data augmentation as its important role in self-supervised learning. Basically, in order to circumvent the randomness in typical augmentation operations, we predetermine all augmentations used in the distillation and record all the corresponding representations of the augmented images for future usage (i.e. the distilled dataset now supports the self-supervised learning approaches which adopt the objectives related to augmented images for training a new model). Although such idea indeed improves the performance, storing all augmented results occupies a lot of memory. To better reduce the memory usage, we propose **approximation networks** which are lightweight and learn to predict the representation shift from an unaugmented distilled image to its augmented views, where in results we only need to store the approximation networks and the representations of unaugmented distilled images.

In detail, given $A$ predetermined augmentations $\{\tau_1, \ldots, \tau_A\}$, the unaugmented distilled images $X_s$ are extended to be $\tilde{X}_s = \{X_s, \tau_1(X_s), \ldots, \tau_A(X_s)\} \in \mathbb{R}^{m(A+1) \times d_x}$, and their corresponding representations $\tilde{Y}_s \in \mathbb{R}^{m(A+1) \times d_y}$ are initialized via $\{(C^y B^y), (C_1^y B^y), \ldots, (C_A^y B^y)\}$ (following the similar idea as described in Section 3.2) where $C_a^y \in \mathbb{R}^{m \times V}$, $a = 1, \ldots, A$, is the coefficient obtained by projecting the representation of distilled images under the $a$-th augmentation, i.e. $g_\phi(\tau_a(X_s))$, onto the bases $B^y$. Furthermore, there are $A$ approximation networks $\{Q_1, \ldots, Q_A\}$ in which the $a$-th network $Q_a : \mathbb{R}^V \to \mathbb{R}^V$ learns to estimate the shift in terms of representation caused by the $a$-th augmentation $\tau_a$. For instance, given an unaugmented distilled image $\tilde{x}_i$ and its augmented view $\tau_a(\tilde{x}_i)$, the difference/shift in terms of their corresponding representations is predicted by $Q_a(c_i^y)$, where $c_i^y$ stands for the parameterized coefficient of $\tilde{y}_i$ (i.e. the representation of $\tilde{x}_i$).

## 3.4 Optimization and Evaluation

The optimization of our proposed self-supervised dataset distillation framework basically follows the bilevel optimization scheme suggested by KRR-ST (Lee et al., 2024) with several modifications to incorporate our techniques of image and representation parameterizations, data augmentation, and feature approximation networks:

1. **Inner loop of bilevel optimization**: The inner model $\hat{g}_\theta$ sampled from the model pool $\mathcal{M}$ is trained with minimizing the inner loss $\mathcal{L}_{\text{inner}}(\theta; \tilde{X}_s, \tilde{Y}_s)$ whose formulation is identical to Equation 2 but takes $(\tilde{X}_s, \tilde{Y}_s)$, i.e. distilled dataset gone through augmentations, as its training dataset.

2. **Outer loop of bilevel optimization**: The outer loop similarly minimizes the outer loss $\mathcal{L}_{\text{outer}}(\tilde{X}_s, \tilde{Y}_s; f_\omega)$ as defined in Equation 3, while the gradients would be further backpropagated to the bases $(B^x, B^y)$ and the coefficients $(C^x, C^y, C_a^y \mid a = 1, \ldots, A)$ in the image and representation spaces, due to our proposed parameterizations (e.g. $X_s = D(C^x B^x, s)$ and $Y_s = C^y B^y$; with noting that the upsampling function $D$ is differentiable). Please note that our predetermined augmentations $\{\tau_1, \ldots, \tau_A\}$ are all differentiable thus do not block the backpropagation through $\tilde{X}_s = \{X_s, \tau_1(X_s), \ldots, \tau_A(X_s)\}$.

3. **After bilevel optimization**: Approximation networks $(Q_a \mid a = 1, \ldots, A)$ are optimized for minimizing the MSE between $Q_a(C^y)$ and $C_a^y - C^y$.

In results, our proposed method stores the image bases $B^x$, image coefficients $C^x$, representation bases $B^y$, representation coefficients $C^y$, and the approximation networks $\{Q_1, \ldots, Q_A\}$. Particularly, our distilled dataset is constructed by $\tilde{X}_s = \{X_s, \tau_1(X_s), \ldots, \tau_A(X_s)\}$ with $X_s = D(C^x B^x, s)$ and their corresponding representations $\tilde{Y}_s^Q = \{Y_s, Y_s + Q_1(C^y)B^y, \ldots, Y_s + Q_A(C^y)B^y\}$ with $Y_s = C^y B^y$. The pseudo-code summary as well as the implementation details of our self-supervised dataset distillation are provided in Supplementary.

As the goal of our distilled dataset $(\tilde{X}_s, \tilde{Y}_s^Q)$ is for further use of training a new model (basically a feature extractor, trained to regress from $\tilde{X}_s$ to $\tilde{Y}_s^Q$) to mimic the characteristics of the self-supervisedly pretrained teacher model $g_\phi$, its evaluation follows the typical linear evaluation scheme of self-supervised learning works: the new model (i.e. feature extractor) learnt from $(\tilde{X}_s, \tilde{Y}_s^Q)$ is frozen and coupled with a linear classifier, where the linear classifier is trained upon the supervised dataset of a downstream task. Higher performance on such downstream task indicates the superior quality of the new feature extractor and consequently the better distilled quality of our dataset $(\tilde{X}_s, \tilde{Y}_s^Q)$.

## 4 EXPERIMENTS

**Datasets.** CIFAR100 (Krizhevsky, 2009), TinyImageNet (Le & Yang, 2015), and ImageNet (Deng et al., 2009) are taken as our source datasets for performing self-supervised DD, while the distilled dataset is evaluated upon the target datasets (which include the source datasets themselves, CIFAR10 (Krizhevsky, 2009), CUB2011 (Wah et al., 2011), and Stanford Dogs (Khosla et al., 2011) for the classification). Noting that we align the image resolution in target datasets with the one in the source dataset (e.g. CIFAR100 uses $32 \times 32$, while both TinyImageNet and ImageNet use $64 \times 64$).

**Baselines.** Several baselines are adopted for making comparison with our method: 1) *Random* randomly draws samples from the source dataset which are couple with their representations extracted by the teacher model to build the distilled dataset; 2) *KMeans* firstly performs kmeans clustering in the feature space of teacher model, where the centroids and their corresponding image samples construct the distilled dataset; 3) four representative approaches of supervised DD, including *DSA* (Zhao & Bilen, 2021a), *DM* (Zhao & Bilen, 2023), *IDM* (Zhao et al., 2023), and *DATM* (Guo et al., 2024); and 4) *KRR-ST* the state-of-the-art method (Lee et al., 2024) of self-supervised dataset distillation. Please note that for the baselines of supervised dataset distillation, the feature extractor used in the linear evaluation is trained by their supervised distilled dataset via cross-entropy loss.

**Storage Budget.** We follow the common practice of dataset distillation (Deng & Russakovsky, 2022; Lee et al., 2024) to adopt the entire memory consumption equivalent to storing $N$ images (where the pixels are stored in floats) as the reference of storage budget. Noting that as our proposed method stores the image/representation bases and coefficients as well as approximation networks, we ensure that the size summation (in floats) of our tensors (for bases and coefficients) and network weights closely approximates the storage budget. Moreover, for supervised distillation baselines, $N$ is the image per class (IPC) times the number of classes in source dataset.

### 4.1 EXPERIMENTAL RESULTS

**Transfer Learning and Cross-Architecture Generalization.** Our experiments investigate the adaptability of our method across datasets and architectures. We highlight the significance of cross-architecture evaluation in determining the quality of distilled datasets. In Table 1, we present a complete comparison of the performance of our method against various baseline approaches across multiple target datasets and network architectures. Results report average accuracy and standard deviation (calculated over three runs). For each experiment, we select or distill samples from CIFAR100, which serve as the source dataset. The storage budget is $N$=100 images (or an equivalent tensor/weight size), which is equivalent to IPC=1 in the conventional supervised dataset distillation setting. For distillation methods, we use 3-layer CNN to distill samples. These samples are used to pretrain a feature extractor, including 3-layer CNN, VGG11 (Simonyan & Zisserman, 2015), ResNet18 (He et al., 2016), AlexNet (Krizhevsky et al., 2012), MobileNet (Howard et al., 2018), Vision Transformer (ViT) (Dosovitskiy et al., 2021). Then the feature extractor undergoes a linear evaluation on target datasets like CIFAR100 (Krizhevsky, 2009), CIFAR10 (Krizhevsky, 2009), CUB2011 (Wah et al., 2011), and Stanford Dogs (Khosla et al., 2011). The result shows that our method effectively distills CIFAR100 into a compact set, which allows to train a generalized feature extractor being compatible for many downstream tasks, where it outperforms all the baselines across various feature extractor architectures and target datasets. The results with adopting TinyImageNet and ImageNet as the source dataset are provided in Supplementary, in which we can draw the consistent observation on our proposed method for providing superior cross-architecture generalization and transfer learning performance (i.e. linear evaluation).

Table 1: The results on various target datasets and feature extractor architectures. We use 3-layer CNN to perform distillation from the CIFAR100 dataset with storage budge 100 images, then a feature extractor are trained on the distilled dataset and linear evaluation is performed on target dataset. We report the average and standard deviation over three runs. The best results are bolded.

| Target | Method | ConvNet | VGG11 | ResNet18 | AlexNet | MobileNet | ViT |
|---|---|---|---|---|---|---|---|
| CIFAR100 | Random | $43.66_{\pm0.57}$ | $23.76_{\pm0.78}$ | $19.26_{\pm0.40}$ | $28.82_{\pm1.17}$ | $11.99_{\pm3.43}$ | $20.70_{\pm0.45}$ |
| | Kmeans | $43.94_{\pm0.34}$ | $25.13_{\pm2.39}$ | $19.05_{\pm0.26}$ | $29.82_{\pm0.73}$ | $11.43_{\pm1.73}$ | $20.77_{\pm0.42}$ |
| | DSA | $39.38_{\pm0.49}$ | $19.97_{\pm2.81}$ | $20.11_{\pm0.09}$ | $31.57_{\pm0.25}$ | $9.58_{\pm0.36}$ | $20.03_{\pm0.19}$ |
| | DM | $31.93_{\pm1.06}$ | $10.36_{\pm0.74}$ | $16.24_{\pm0.45}$ | $20.49_{\pm0.27}$ | $8.06_{\pm1.00}$ | NaN |
| | IDM | $38.71_{\pm0.70}$ | $14.24_{\pm0.88}$ | $19.05_{\pm0.04}$ | $33.71_{\pm0.31}$ | $8.18_{\pm0.46}$ | $17.41_{\pm0.64}$ |
| | DATM | $38.73_{\pm0.31}$ | $26.04_{\pm1.11}$ | $\mathbf{21.20}_{\pm0.34}$ | $29.31_{\pm0.35}$ | $10.17_{\pm1.05}$ | $20.11_{\pm0.44}$ |
| | KRR-ST | $47.00_{\pm0.51}$ | $27.78_{\pm0.84}$ | $18.92_{\pm0.27}$ | $31.27_{\pm0.57}$ | $10.11_{\pm1.60}$ | $20.82_{\pm0.22}$ |
| | Ours | $\mathbf{52.41}_{\pm0.10}$ | $\mathbf{35.35}_{\pm0.37}$ | $20.90_{\pm0.71}$ | $\mathbf{36.88}_{\pm0.37}$ | $\mathbf{24.14}_{\pm1.46}$ | $\mathbf{23.33}_{\pm0.06}$ |
| CIFAR10 | Random | $68.56_{\pm0.13}$ | $50.23_{\pm2.47}$ | $42.71_{\pm0.26}$ | $52.46_{\pm0.94}$ | $34.95_{\pm1.66}$ | $45.02_{\pm0.23}$ |
| | Kmeans | $67.71_{\pm0.39}$ | $50.59_{\pm1.08}$ | $42.34_{\pm0.63}$ | $53.75_{\pm0.95}$ | $33.53_{\pm2.44}$ | $45.05_{\pm0.62}$ |
| | DSA | $65.67_{\pm0.83}$ | $39.58_{\pm5.70}$ | $44.47_{\pm0.82}$ | $54.21_{\pm0.64}$ | $30.93_{\pm2.32}$ | $42.49_{\pm0.49}$ |
| | DM | $56.27_{\pm1.91}$ | $28.37_{\pm2.27}$ | $42.71_{\pm0.41}$ | $43.12_{\pm1.23}$ | $32.57_{\pm1.63}$ | NaN |
| | IDM | $64.45_{\pm0.37}$ | $34.43_{\pm2.94}$ | $44.77_{\pm0.26}$ | $57.71_{\pm0.35}$ | $31.01_{\pm1.58}$ | $39.34_{\pm1.11}$ |
| | DATM | $66.17_{\pm0.68}$ | $46.95_{\pm0.90}$ | $\mathbf{45.28}_{\pm0.18}$ | $53.69_{\pm0.55}$ | $29.05_{\pm0.56}$ | $43.24_{\pm0.47}$ |
| | KRR-ST | $72.14_{\pm0.60}$ | $51.96_{\pm1.22}$ | $43.37_{\pm0.71}$ | $56.24_{\pm1.75}$ | $34.44_{\pm1.51}$ | $44.90_{\pm0.34}$ |
| | Ours | $\mathbf{76.83}_{\pm0.18}$ | $\mathbf{59.60}_{\pm1.01}$ | $44.12_{\pm0.55}$ | $\mathbf{62.45}_{\pm0.08}$ | $\mathbf{48.73}_{\pm0.63}$ | $\mathbf{47.06}_{\pm0.36}$ |
| CUB2011 | Random | $8.88_{\pm0.56}$ | $5.40_{\pm0.61}$ | $2.71_{\pm0.10}$ | $7.46_{\pm0.23}$ | $1.82_{\pm0.03}$ | $2.69_{\pm0.09}$ |
| | Kmeans | $9.41_{\pm0.07}$ | $5.92_{\pm0.65}$ | $2.76_{\pm0.16}$ | $7.61_{\pm0.19}$ | $2.04_{\pm0.11}$ | $2.73_{\pm0.14}$ |
| | DSA | $6.89_{\pm0.27}$ | $3.70_{\pm0.68}$ | $2.56_{\pm0.25}$ | $6.22_{\pm0.28}$ | $1.57_{\pm0.09}$ | $2.59_{\pm0.10}$ |
| | DM | $5.84_{\pm0.29}$ | $1.74_{\pm0.06}$ | $2.09_{\pm0.05}$ | $2.93_{\pm1.07}$ | $1.25_{\pm0.17}$ | NaN |
| | IDM | $7.09_{\pm0.17}$ | $3.05_{\pm0.17}$ | $2.56_{\pm0.11}$ | $7.01_{\pm0.07}$ | $1.39_{\pm0.19}$ | $2.12_{\pm0.25}$ |
| | DATM | $6.73_{\pm0.15}$ | $5.88_{\pm0.11}$ | $2.64_{\pm0.04}$ | $6.70_{\pm0.38}$ | $1.32_{\pm0.25}$ | $2.47_{\pm0.53}$ |
| | KRR-ST | $10.43_{\pm0.28}$ | $6.58_{\pm0.49}$ | $2.91_{\pm0.12}$ | $8.09_{\pm0.37}$ | $1.88_{\pm0.17}$ | $2.74_{\pm0.22}$ |
| | Ours | $\mathbf{12.24}_{\pm0.65}$ | $\mathbf{8.92}_{\pm0.19}$ | $\mathbf{3.59}_{\pm0.25}$ | $\mathbf{8.27}_{\pm0.23}$ | $\mathbf{4.37}_{\pm0.19}$ | $\mathbf{3.99}_{\pm0.07}$ |
| Stanford Dogs | Random | $12.18_{\pm0.26}$ | $6.48_{\pm1.24}$ | $4.42_{\pm0.09}$ | $8.42_{\pm0.29}$ | $2.82_{\pm0.20}$ | $3.75_{\pm0.23}$ |
| | Kmeans | $12.36_{\pm0.23}$ | $6.48_{\pm0.45}$ | $4.29_{\pm0.29}$ | $8.03_{\pm0.22}$ | $2.52_{\pm0.14}$ | $3.97_{\pm0.07}$ |
| | DSA | $9.97_{\pm0.12}$ | $6.46_{\pm0.34}$ | $4.81_{\pm0.21}$ | $7.93_{\pm0.22}$ | $2.17_{\pm0.07}$ | $3.66_{\pm0.06}$ |
| | DM | $7.20_{\pm0.19}$ | $2.84_{\pm0.32}$ | $3.34_{\pm0.04}$ | $4.26_{\pm0.11}$ | $2.17_{\pm0.35}$ | NaN |
| | IDM | $9.56_{\pm0.13}$ | $4.40_{\pm1.07}$ | $4.22_{\pm0.42}$ | $8.69_{\pm0.15}$ | $1.61_{\pm0.07}$ | $3.78_{\pm0.21}$ |
| | DATM | $9.70_{\pm0.21}$ | $6.32_{\pm0.25}$ | $5.00_{\pm0.15}$ | $7.65_{\pm0.35}$ | $1.90_{\pm0.09}$ | $3.86_{\pm0.19}$ |
| | KRR-ST | $13.42_{\pm0.09}$ | $7.84_{\pm0.50}$ | $4.13_{\pm0.28}$ | $8.42_{\pm0.47}$ | $2.37_{\pm0.07}$ | $3.88_{\pm0.16}$ |
| | Ours | $\mathbf{15.34}_{\pm0.28}$ | $\mathbf{9.36}_{\pm0.22}$ | $\mathbf{5.01}_{\pm0.12}$ | $\mathbf{8.87}_{\pm0.09}$ | $\mathbf{5.48}_{\pm0.26}$ | $\mathbf{5.06}_{\pm0.24}$ |

**Storage Budget Size.** We conduct an experiment to check the impact under variant storage budgets. In this experiment, CIFAR100 is used as source dataset and target dataset simutaneously. For Random and KMeans baselines, we get the same numbers of coreset images as the given budget. For distillation, we use 3-layer CNN to find the distilled dataset from the given dataset within the storage budget. These coresets or distilled datasets are used to pretrain a 3-layer CNN, then linear evaluations for the pretrained CNN are conducted on the given dataset. As the result, Table 2 shows the linear evaluation performance. The proposed method outperforms other baselines in various memory budgets, while two supervised methods, i.e. DSA and IDM, do not scale up as the budget size increases. For comparison, we pretrained the 3-layer CNN using the full CIFAR-100 training dataset as the upperbound for linear evaluation, achieving a result of $59.54\%$. Notably, supervised methods require a memory budget of at least 1 IPC.

**Ablation Study for Our Proposed Methods.** We conduct a study to evaluate the impact of key components, i.e. image and representation parameterization (cf. Section 3.2) as well as predefined augmentation and approximation networks (cf. Section 3.3), introduced in our methodology, where the results are provided in Table 3. Utilizing CIFAR100 for both source and target datasets, our experiments are conducted under a constrained storage budget of $N = 100$ in this study. The baseline for our analysis is established by the KRR-ST (Lee et al., 2024) method, whose accuracy is

Table 2: Linear evaluation results on CIFAR100 with various memory budget $N$. In this experiment, the given dataset is serving as source and target dataset at the same time. We report the average and standard deviation on three runs.

| Memory Budget $N$ | 25 | 50 | 100 | 1000 | 5000 |
|---|---|---|---|---|---|
| Random | $41.61_{\pm 0.45}$ | $41.80_{\pm 0.33}$ | $43.66_{\pm 0.57}$ | $49.82_{\pm 0.62}$ | $52.76_{\pm 0.80}$ |
| KMeans | $39.96_{\pm 0.93}$ | $41.68_{\pm 0.59}$ | $43.94_{\pm 0.34}$ | $49.77_{\pm 0.37}$ | $52.80_{\pm 0.76}$ |
| DSA | - | - | $39.38_{\pm 0.49}$ | $36.16_{\pm 0.80}$ | $36.25_{\pm 0.61}$ |
| DM | - | - | $31.93_{\pm 1.06}$ | $34.96_{\pm 2.92}$ | $38.76_{\pm 1.85}$ |
| IDM | - | - | $38.71_{\pm 0.70}$ | $37.21_{\pm 0.89}$ | $42.19_{\pm 1.02}$ |
| DATM | - | - | $38.73_{\pm 0.31}$ | $44.98_{\pm 0.27}$ | $46.23_{\pm 0.26}$ |
| KRR-ST | $44.06_{\pm 0.41}$ | $45.79_{\pm 0.27}$ | $47.00_{\pm 0.51}$ | $51.89_{\pm 0.21}$ | $52.49_{\pm 0.77}$ |
| Ours | $\mathbf{51.41}_{\pm 0.47}$ | $\mathbf{52.08}_{\pm 0.08}$ | $\mathbf{52.41}_{\pm 0.10}$ | $\mathbf{53.54}_{\pm 0.58}$ | $\mathbf{55.53}_{\pm 0.64}$ |

Table 3: Ablation study on CIFAR100 with memory budget $N$=100. All our proposed techniques contribute to the improvement.

| | Accuracy |
|---|---|
| Baseline (KRR-ST) | $47.00_{\pm 0.51}$ |
| + Parameterization | $48.57_{\pm 0.18}$ |
| + Aug. & Approx. | $52.41_{\pm 0.10}$ |

Table 4: Performance variation caused by initialization for the parameterization (experiments conducted on CIFAR100 with $N$=100).

| | Basis init. | Coeff. init. | Accuracy |
|---|---|---|---|
| I) | Random | Random | $22.05_{\pm 3.04}$ |
| II) | Real | Random | $30.99_{\pm 0.25}$ |
| III) | PC | Projection | $52.41_{\pm 0.10}$ |

$47.00\%$. Upon integrating the "Image and Representation Parameterization" technique (+Parameterization), we observe an improvement to $48.57\%$ in accuracy. Further enhancements are achieved by incorporating "Augmentation and Feature Approximation" (+Aug. & Approx.), leading to a notable improvement to $52.41\%$. These results reveal that both our proposed techniques significantly contribute to the performance of self-supervised dataset distillation.

**Different Initialization Methods for Bases and Coefficients.** In Section 3.2 and Section 3.3 we have mentioned the way of initializing the bases and coefficients used in our proposed method. Here we conduct experiments to study the impact of different initializations upon the distillation performance. Basically, we test three methods, denoted as I), II), and III) respectively, to initialize the bases and coefficients for images and representations, where the experiments are carried out on CIFAR100 which serves as both source and target datasets with a storage budget $N = 100$. I) Both the bases and the coefficients are initialized randomly by a standard Gaussian distribution. In this setting, the final distillation dataset only trains a feature extractor with linear evaluation precision $22.05\%$; II) The bases are simply initialized by random images drawn from the source dataset, and the coefficients are initialized randomly by a standard Gaussian distribution. This method obtains $30.09\%$ accuracy; III) Our proposed method utilizes principal components (denoted as PC) of the source dataset to initialize the image bases, while the representation bases are initialized by the principal components of the source data representations extracted by the teacher model. It can get $52.41\%$ accuracy. As in Table 4, this experiment shows that the initialization of bases and coefficients is critical, and different initialization methods would cause up to $30\%$ loss on the performance.

## 5 Conclusion

Our work introduces an novel self-supervised dataset distillation method that reduces the size of training datasets while preserving model performance across various architectures. Our method leverages the parameterization of distilled images and representations into bases and coefficients, along with predefined augmentations to prevent gradient bias caused by the randomness in the bilevel optimization. Additionally, we employ approximation networks to capture the relationships between different augmentations, further reducing storage costs. Experiments demonstrate the superior linear evaluation results of the feature extractor pretrained on our distilled datasets across various target datasets and architectures, emphasizing the compactness and generalizability of our approach.

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

## A APPENDIX

### A.1 IMPLEMENTATION DETAILS

In adherence to the methodology described in KRR-ST (Lee et al., 2024), the inner model adopted in our approach utilizes convolutional layers that include batch normalization (Ioffe & Szegedy, 2015), ReLU activation, and average pooling. The number of layers of inner model is determined by the image resolution, adopting 3 layers for images sized $32 \times 32$ and 4 layers for $64 \times 64$ images. The model pool for inner models (cf. the last paragraph of Section 3.1), denoted as $\mathcal{M}$, consists of 10 models, which are initialized and updated via full-batch gradient descent, with learning rate and momentum set to $0.1$, $0.9$, respectively. The update steps $Z$ are $1,000$. To optimize our distilled dataset, we employ the AdamW optimizer (Loshchilov & Hutter, 2019), starting with a learning rate of $0.001$ that linearly decayed. This distillation process involves $30,000$ outer iterations for CIFAR100 and $20,000$ for TinyImageNet and ImageNet. The ResNet18 model (He et al., 2016) is serving as a self-supervised teacher $g_\phi$ and is trained with the Barlow Twins objective (Zbontar et al., 2021) (where the training is based on the solo-learn library (da Costa et al., 2022)). Our settings of the number of image bases $U$ and the number of representation bases $V$ are listed in the Table 5. Notably, the number of bases is generally set to be twice the memory budget, $N$, except in cases where this exceeds the dimensional limits of the images or representations. We set the size of image basis to $3 \times 16 \times 16$ and the size of representation basis to $512$. We adopt bilinear upsampling as our upsampling function $D(\cdot)$ and and the upsampling scale is set to 2 for CIFAR100 and 4 for TinyImageNet and ImageNet. We use image rotation as our augmentation function, which rotates the image by $90°$, $180°$ and $270°$. The approximation networks are designed as a 2-layer perceptron, with the hidden layer sizes being 4 for CIFAR100 and 16 for TinyImageNet and ImageNet.

Upon completion of the distillation process and stepping forward to evaluation, we pretrain a model (which is acting as feature extractor later for linear evaluation) on the distilled dataset for $1,000$ epochs. This pretraining employs a stochastic gradient descent (SGD) optimizer with a mini-batch size of $256$, where the learning rate and momentum are maintained at $0.1$ and $0.9$, respectively. The weight decay parameters during pretraining the feature extractor are listed in Table 5, we set the weight decay parameters which depends on the size of distilled dataset. For training the linear classifier to conduct linear evaluation, we standardize the experimental settings to utilize the SGD optimizer with a momentum of $0.9$, excluding weight decay, and initiate the learning rate of task-specific head to $0.2$ with cosine scheduling.

In Table 14, we provide a detailed list of all the notations/variables used in our description and their corresponding meanings.

Table 5: Hyper-parameter configurations for our experiments. The storage budget is allocated to equivalently store $N$ images, while the numbers of image bases and representation bases are set to $U$ and $V$, respectively.

| Dataset | $N$ | $U$ | $V$ | Weight Decay |
|---------|-----|-----|-----|--------------|
| CIFAR100 | 25 | 50 | 50 | 0.001 |
| CIFAR100 | 50 | 100 | 100 | 0.001 |
| CIFAR100 | 100 | 200 | 200 | 0.001 |
| CIFAR100 | 1000 | 500 | 500 | 0.0001 |
| CIFAR100 | 5000 | 700 | 500 | 0.0001 |
| TinyImageNet | 50 | 100 | 100 | 0.005 |
| TinyImageNet | 100 | 200 | 200 | 0.005 |
| TinyImageNet | 200 | 400 | 400 | 0.005 |
| ImageNet | 250 | 500 | 500 | 0.005 |

### A.2 PSEUDO CODE

Here we provide the pseudo code (i.e. Algorithm 1) together with the detailed but compact explanation to emphasize the systematic approach of our proposed method for self-supervised dataset distillation, which begins with initializing the framework and proceeds through a bilevel optimization process, ending with the training of approximation networks to capture representation shifts due to the augmentations (e.g. rotations). Such pseudo code offers a clear understanding of our method's structure and objectives.

The initialization phase, from Line 1 to Line 6, starts with training a teacher model using a self-supervised learning (SSL) objective on an unlabeled dataset (i.e. the source dataset that we are about to distill). It involves setting up the bases and coefficients for the image and representation, following the guidelines detailed in Section 3.2. This phase also includes the initialization of a pool of feature extractor models, each associated with a specific training iteration drawn from a predefined range, and the models are trained on the initial distilled images and their corresponding targets through the process defined in Algorithm 2.

The core of the Algorithm 1, spanning from Lines 7 to 20, engages in bilevel optimization. It involves iteratively refining the distilled images and representations by adjusting their bases and coefficients. First, it randomly selects a model from the pool to update the bases and coefficients in order to minimize the outer loss, as shown in Equation 3. This step also includes an inner loop where the selected model is further trained using Equation 2 if its assigned steps are below the maximum limit; otherwise, it is reinitialized.

In the final stage (from Lines 21 to 23), the algorithm addresses the shifts in representations resulting from our predefined augmentations. This is achieved by training approximation networks to minimize the mean squared error (MSE) between the predicted coefficient shifts and the difference in coefficients caused by augmentations. The distilled dataset is thus represented by the bases, coefficients, and trained approximation networks.

### A.3 TRANSFER LEARNING AND CROSS-ARCHITECTURE GENERALIZATION ON TINYIMAGENET

Our experiments explore the versatility of our method across different datasets and architectures. We emphasize the importance of evaluating across architectures to assess the efficacy of distilled datasets. In Table 6, we offer a comprehensive comparison of performance against several baseline approaches across a variety of target datasets and network architectures. The results include average accuracy and standard deviation, based on three runs. For each experiment, we select or distill samples from TinyImageNet, which serve as the source dataset. The storage budget is $N$=200 images (or an equivalent tensor/weight size), which corresponds to Image Per Class (IPC) of 1 as

---

**Algorithm 1** The proposed self-supervised dataset distillation

---

**Input:** Unlabeled dataset $X_t$, the number of image bases $U$, the number of representation bases $V$, upsampling function $D$, scaling factor $s$, predefined augmentations $\{\tau_a\}_{a=1}^A$, total steps $Z$
1: Optimize a teacher model $g_\phi$ with the SSL objective on $X_t$.
2: Initialize bases and coefficients $B^x, B^y, C^x, C^y, \{C_a^y\}_{a=1}^A$. (c.f. Section 3.2)
3: Get $\tilde{X}_s, \tilde{Y}_s$ from Algorithm 2.
4: Randomly initialize models $\hat{g}_{\theta_l}$ and integers $z_l \in \{1, \ldots, Z\}$ for $l = 1, \ldots, L$
5: Train model $\hat{g}_{\theta_l}$ for $z_l$ steps on $\tilde{X}_s$ and $\tilde{Y}_s$ for $l = 1, \ldots, L$.
6: Initialize model pool $\mathcal{M} \leftarrow \{(\hat{g}_{\theta_1}, z_1), \ldots, (\hat{g}_{\theta_L}, z_L)\}$.
7: **for** each distillation step **do**
8:      Get $\tilde{X}_s, \tilde{Y}_s$ from Algorithm 2.
9:      Sample a model $\hat{g}_{\theta_j} = h_{W_j} \circ f_{\omega_j}$, and its trained steps $z_j$ from $\mathcal{M}$.
10:      Compute the outer objective $\mathcal{L}_{\text{outer}}(\tilde{X}_s, \tilde{Y}_s; f_{\omega_j})$ using Equation 3.
11:      Get the gradient $\nabla \mathcal{L}_{\text{outer}}$ w.r.t. $B^x, B^y, C^x, C^y$, and $C_a^y$ for $a = 1, \ldots, A$.
12:      Update bases and coefficients, $B^x, B^y, C^x, C^y$, and $C_a^y$ for $a = 1, \ldots, A$.
13:      **if** $z_j < Z$ **then**
14:          Set $z_j \leftarrow z_j + 1$
15:          Evaluate inner loss $\mathcal{L}_{\text{inner}}(\theta_j; \tilde{X}_s, \tilde{Y}_s)$, using Equation 2.
16:          Update $\theta_j$ according to $\nabla \mathcal{L}_{\text{inner}}(\theta_j; \tilde{X}_s, \tilde{Y}_s)$.
17:      **else**
18:          Reset $z_j \leftarrow 0$ and randomly initialize $\theta_j$
19:      **end if**
20: **end for**
21: Randomly initialize approximation networks $\{Q_1, \ldots, Q_A\}$
22: Get the representation shift, $C_a^y - C^y$ for $a = 1, \ldots, A$
23: Train $Q_a$ by minimizing MSE between $Q_a(C^y)$ and $C_a^y - C^y$ for $a = 1, \ldots, A$
**Output: distilled data** $B^x, B^y, C^x, C^y, \{Q_1, \ldots, Q_A\}$**.**

---

**Algorithm 2** Generate distilled images and target representations

---

**Input:** image bases $B^x$, image coefficients $C^x$, representation bases $B^y$, representation coefficients $C^y$, upsampling function $D$, scaling factor $s$, predefined augmentations $\{\tau_a\}_{a=1}^A$
1: Generate distilled images $X_s \leftarrow D(C^x B^x, s)$.
2: Generate augmented images $\tilde{X}_s \leftarrow \{X_s, \tau_1(X_s), \ldots, \tau_A(X_s)\}$.
3: Generate target representations on $\tilde{Y}_s \leftarrow \{(C^y B^y), (C_1^y B^y), \ldots, (C_A^y B^y)\}$.
**Output:** $\tilde{X}_s$**,** $\tilde{Y}_s$**.**

---

typically defined in the supervised dataset distillation setting. We employ a 4-layer Convolutional Neural Network for the distillation process. These distilled samples are then utilized to pretrain a feature extractor, which includes architectures such as a 4-layer CNN, VGG11 (Simonyan & Zisserman, 2015), ResNet18 (He et al., 2016), AlexNet (Krizhevsky et al., 2012), and MobileNet (Howard et al., 2018). Subsequently, we perform a linear evaluation to assess performance of the feature extractor on target datasets like TinyImageNet (Le & Yang, 2015), CUB2011 (Wah et al., 2011), and Stanford Dogs (Khosla et al., 2011). The findings demonstrate that our method efficiently condenses TinyImageNet into a compact distilled dataset. This enables the training of a versatile feature extractor that performs better than baselines in various downstream tasks across various feature extractor architectures and target datasets.

Table 6: The results on various target datasets and feature extractor architectures. We use 4-layer CNN to perform distillation from the TinyImageNet dataset with storage budget 200 images, then a feature extractor are trained on the distilled dataset and linear evaluation is performed on target dataset. We report the average and standard deviation over three runs. The best results are bolded.

| Target | Method | ConvNet | VGG11 | ResNet18 | AlexNet | MobileNet |
|---|---|---|---|---|---|---|
| TinyImageNet | Random | $24.44_{\pm 0.5}$ | $11.71_{\pm 1.66}$ | $14.11_{\pm 0.71}$ | $9.84_{\pm 1.13}$ | $6.51_{\pm 0.3}$ |
| | KRR-ST | $28.54_{\pm 0.47}$ | $14.66_{\pm 1.17}$ | $14.78_{\pm 0.36}$ | $10.28_{\pm 1.44}$ | $6.19_{\pm 0.48}$ |
| | Ours | $\mathbf{29.54}_{\pm 0.26}$ | $\mathbf{25.12}_{\pm 0.27}$ | $\mathbf{16.75}_{\pm 0.56}$ | $\mathbf{23.28}_{\pm 0.15}$ | $\mathbf{14.02}_{\pm 0.29}$ |
| CUB2011 | Random | $9.37_{\pm 0.19}$ | $7.95_{\pm 0.78}$ | $3.25_{\pm 0.09}$ | $5.02_{\pm 1.52}$ | $2.17_{\pm 0.23}$ |
| | KRR-ST | $\mathbf{11.59}_{\pm 0.35}$ | $8.27_{\pm 0.67}$ | $3.70_{\pm 0.17}$ | $5.24_{\pm 0.09}$ | $2.17_{\pm 0.38}$ |
| | Ours | $11.11_{\pm 0.26}$ | $\mathbf{10.58}_{\pm 0.30}$ | $\mathbf{5.17}_{\pm 0.19}$ | $\mathbf{9.78}_{\pm 0.16}$ | $\mathbf{5.78}_{\pm 0.19}$ |
| Stanford Dogs | Random | $12.38_{\pm 0.25}$ | $7.41_{\pm 0.95}$ | $4.79_{\pm 0.14}$ | $5.99_{\pm 0.94}$ | $2.97_{\pm 0.07}$ |
| | KRR-ST | $14.52_{\pm 0.17}$ | $8.79_{\pm 1.02}$ | $4.61_{\pm 0.23}$ | $5.49_{\pm 0.36}$ | $2.48_{\pm 0.06}$ |
| | Ours | $\mathbf{14.62}_{\pm 0.26}$ | $\mathbf{12.29}_{\pm 0.07}$ | $\mathbf{6.60}_{\pm 0.06}$ | $\mathbf{11.30}_{\pm 0.21}$ | $\mathbf{7.21}_{\pm 0.54}$ |

## A.4 VARY THE STORAGE BUDGET ON TINYIMAGENET

We examine the impact of different storage budgets. In this experiment, TinyImageNet serves as both the source and target dataset. For the Random and KMeans baselines, we select the same number of coreset images as allowed by the given budget. For the distillation process, a 4-layer CNN is employed to extract a distilled dataset from the original data within the specified storage budget. These coresets or distilled datasets are then used to pretrain a 4-layer CNN, followed by linear evaluations on the dataset. As shown in Table 7, the linear evaluation results indicate that our proposed method surpasses other baselines across various memory budgets.

Table 7: Linear evaluation results on TinyImageNet (Le & Yang, 2015) with various memory budget $N$. In this experiment, the given dataset is serving as source and target dataset at the same time. We report the average and standard deviation on three runs.

| Memory Budget $N$ | 50 | 100 | 200 |
|---|---|---|---|
| Random | $22.43_{\pm 0.54}$ | $22.73_{\pm 0.35}$ | $23.95_{\pm 0.30}$ |
| KMeans | $23.17_{\pm 0.23}$ | $23.96_{\pm 0.21}$ | $25.03_{\pm 0.34}$ |
| KRR-ST | $25.29_{\pm 0.30}$ | $25.64_{\pm 0.19}$ | $27.23_{\pm 0.17}$ |
| Ours | $\mathbf{28.03}_{\pm 0.11}$ | $\mathbf{29.35}_{\pm 0.48}$ | $\mathbf{31.25}_{\pm 0.17}$ |

## A.5 TRANSFER LEARNING AND CROSS-ARCHITECTURE GENERALIZATION ON IMAGENET

In this experiment, ImageNet is used as the source dataset, with a storage budget of $N = 250$ images (or an equivalent tensor/weight size). For the Random baseline, we select the same number of images as permitted by the budget. Following the settings of KRR-ST (Lee et al., 2024), a 4-layer CNN is used during the distillation process to generate a distilled dataset with a resolution of $64 \times 64$ within the specified storage constraints. These coresets or distilled datasets are then used to pretrain a

feature extractor with architectures such as a 4-layer CNN, VGG11 (Simonyan & Zisserman, 2015), ResNet18 (He et al., 2016), AlexNet (Krizhevsky et al., 2012), and MobileNet (Howard et al., 2018). Linear evaluations are subsequently performed on target datasets like TinyImageNet (Le & Yang, 2015), CUB2011 (Wah et al., 2011), and Stanford Dogs (Khosla et al., 2011). As shown in Table 8, the linear evaluation results demonstrate that our proposed method consistently outperforms other baselines in various downstream tasks across different backbone architectures.

Table 8: The linear evaluation results on various target datasets and feature extractor architectures. We use 4-layer CNN to perform distillation from the ImageNet (Deng et al., 2009) dataset with storage budge $N$=250 images, then feature extractors are trained on the distilled dataset and linear evaluation are performed on various target dataset. The best results are bolded.

| | | ConvNet | VGG11 | ResNet18 | AlexNet | Mobilenet |
|---|---|---|---|---|---|---|
| | Random | 13.65 | 6.59 | 5.79 | 5.87 | 2.45 |
| ImageNet | KRR-ST | 16.75 | 10.22 | 6.19 | 6.81 | 3.18 |
| | Ours | **21.17** | **14.8** | **8.04** | **11.31** | **10.72** |
| | Random | 25.44 | 10.51 | 13.84 | 9.12 | 7.28 |
| TinyImageNet | KRR-ST | 29.17 | 19.03 | 13.74 | 8.71 | 7.84 |
| | Ours | **33.04** | **27.23** | **18.89** | **18.83** | **22.35** |
| | Random | 10.65 | 7.56 | 3.61 | 4.90 | 2.66 |
| CUB2011 | KRR-ST | **12.32** | 10.44 | 3.78 | 4.76 | 2.38 |
| | Ours | 12.08 | **11.70** | **6.04** | **10.87** | **9.10** |
| | Random | 13.16 | 7.04 | 4.97 | 6.27 | 2.93 |
| Stanford Dogs | KRR-ST | 15.90 | 10.62 | 4.86 | 5.87 | 3.67 |
| | Ours | **17.2** | **12.80** | **7.83** | **11.74** | **10.84** |

## A.6 RESULTS ON IMAGENET WITH LARGER IMAGE RESOLUTIONS

**ImageNette ($128 \times 128$).** We evaluated our method on higher-resolution data using ImageNette, a 10-class subset of ImageNet, as both the source and target dataset. We set the resolution to $128 \times 128$ and used a storage budget of 10 images. For the distillation process, we employed a 5-layer CNN to generate distilled samples. These samples were then used to pretrain a randomly initialized 5-layer CNN feature extractor, and we report the linear evaluation accuracy on this feature extractor. As shown in Table 9, our method scales effectively to larger resolutions.

**ImageNet-1K ($224 \times 224$).** We further tested our method on ImageNet-1K at a resolution of $224 \times 224$ with a storage budget of 100 images. For distillation process, we use a 6-layer CNN to create distilled samples. These samples are used to pretrain a random initialized 6-layer CNN feature extractor, and we report the linear evaluation accuracy which is conducted on this feature extractor. The results, showed in Table 10, demonstrate that our method surpass baseline methods on larger scale datasets.

Table 9: Linear evaluation accuracy on a 5-layer ConvNet pretrained with 10 distilled ImageNette images ($128 \times 128$).

| Method | Accuracy |
|---|---|
| Random | 46.93 |
| KRR-ST | 50.14 |
| Ours | **59.31** |

Table 10: Linear evaluation accuracy on a 6-layer ConvNet pretrained with 100 distilled ImageNet-1K images ($224 \times 224$).

| Method | Accuracy |
|---|---|
| Random | 8.75 |
| KRR-ST | 9.22 |
| Ours | **9.60** |

## A.7 MODELING THE REPRESENTATION SHIFT CAUSED BY AUGMENTATION

In this study, we explore the impact of different methods in predicting the representation shift through an experiment. Using CIFAR100 (Krizhevsky, 2009) as both the source and target dataset

with storage budget $N = 100$, we examine three distinct scenarios to model this phenomenon. The first scenario, termed "Same," involves treating the representation of all augmented images as identical, a technique commonly employed in self-supervised learning. This approach typically regards augmented views generated from a single image as a positive pair, aiming to align their representations closely. The second scenario, "Bias," presupposes that the augmented view diverges from the original view by a specific bias vector, with each predefined augmentation associated with its unique bias vector. The third scenario, "Ours," is described in our main paper. It adopts approximation networks to predict the representation shifts, and the subindex 2, 4, and 8 indicate the width of the hidden layer in the approximation networks. Last, "Ideal", which serves as an upperbound in this experiment, represents storing all of the representation without considering the storage budget. According to the results presented in Table 11, including linear evaluation accuracy of the feature extractor trained on distilled data and prediction mean square error (MSE), the proposed lightweight approximation networks can achieve better accuracy than "Same" and "Bias" baselines. Notably, while larger networks achieve lower MSE, they do not always improve accuracy due to the storage budget constraint. These findings indicate that the proposed design can effectively predict representation shifts caused by augmentations, achieving a reasonable trade-off between MSE and accuracy.

Table 11: Ablation study on modeling the target of augmented images. The result shows the linear evaluation accuracy which is conducted on a 3-layer ConvNet. It is pretrained on the distilled CIFAR100 with $N$=100.

| Method | Same | Bias | $\text{Ours}_2$ | $\text{Ours}_4$ | $\text{Ours}_8$ | Ideal |
|---|---|---|---|---|---|---|
| Accuracy | 50.10 | 50.32 | 52.30 | **52.41** | 51.19 | 53.51 |
| MSE | 0.31 | 0.30 | 0.07 | 0.06 | 0.04 | - |

## A.8 AUGMENTATION.

Our investigation delves into the effects of various predefined augmentations applied within our proposed method, utilizing the CIFAR100 as both source and target dataset. We specifically explore three distinct augmentations: rotation, jigsaw, and crop, each of which is differentiable and yields a predetermined (i.e., non-random) result. For the rotation augmentation, we subject an image to rotations of $90°, 180°$, and $270°$. The jigsaw augmentation involves dividing an image into four patches. We then perform augmentation by swapping the patches in three ways: left with right, and top with bottom, as well as a combination of both swaps. In the crop augmentation process, an image is cropped into its four corners and a central portion, with each cropped area being $20 \times 20$ pixels. These cropped sections are subsequently resized to the original resolution of the images. The result of the experiment is shown in Table 12, all of them are better than no augmentation, indicating that our "Predefined Data Augmentation and Feature Approximation" is not sensitive to the choice of augmentation, while adopting rotation augmentations achieves the best.

Table 12: Study on choices of predefined augmentations (dataset: CIFAR100; $N$=100).

| No augmentation | Roatation | Jigsaw | Crop |
|---|---|---|---|
| $48.57_{\pm 0.18}$ | $52.41_{\pm 0.10}$ | $50.93_{\pm 0.21}$ | $51.03_{\pm 0.40}$ |

## A.9 VISUALIZATION

We conduct a qualitative analysis of our outcomes by distilling CIFAR100 and TinyImageNet within a storage budget $N = 100$ and 200, respectively. First, we show the bases which are initialized by the first 64 largest principal components in Figure 2a. To illustrate the distilled images, we combine the image bases with their coefficients into distilled images and present a subset of these images, as depicted in Figure 2b. Additionally, we conduct a comparative analysis between the CIFAR100 representations extracted by the teacher model and those obtained from our distillation process, which are parameterized by the combination of representation bases and coefficients. For

this purpose, we employ the UMAP method, as detailed in (McInnes et al., 2018), to project both the CIFAR100 representations and the distilled target representations into two-dimensional vectors. The resultant visualization is shown in Figure 2c, where we can see that the distilled data have the similar distribution as real dataset, indicating that the distilled data captures the characteristics of the real dataset. Finally, we perform the same visualization on TinyImageNet, and Figures 3a, 3b, and 3c show similar visualization results.

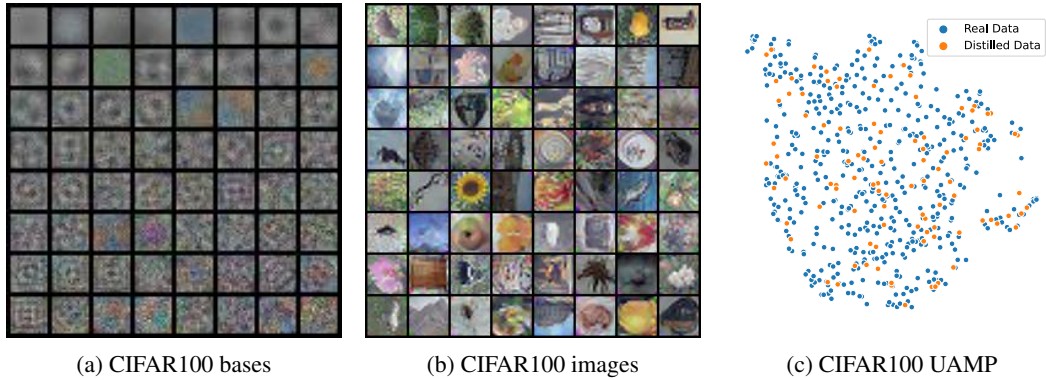

|     (a) CIFAR100 bases     |     (b) CIFAR100 images     |     (c) CIFAR100 UAMP     |

Figure 2: Visualization result of CIFAR100 ($N = 100$)

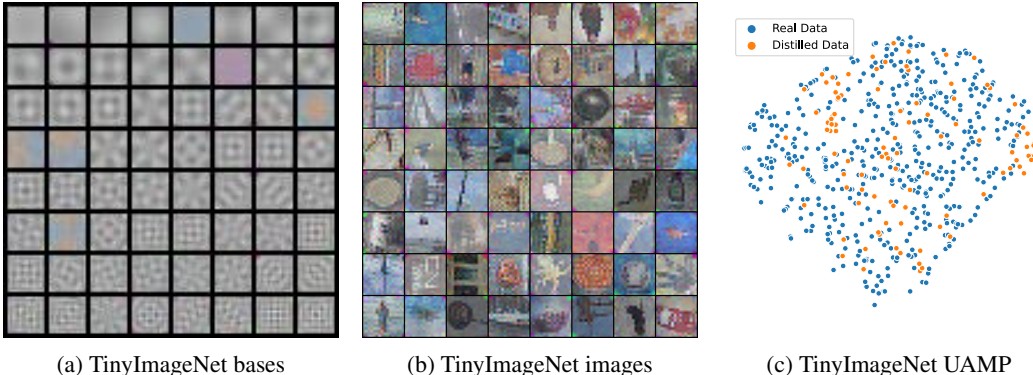

|   (a) TinyImageNet bases   |   (b) TinyImageNet images   |   (c) TinyImageNet UAMP   |

Figure 3: Visualization result of TinyImageNet ($N = 200$)

### A.10    COMPUTATIONAL COST

Based on CIFAR-100 with storage buffer $N = 100$, we evaluated GPU memory usage and execution time. The results demonstrate that while our method requires approximately 1.5x more GPU memory than KRR-ST, it remains feasible within modern hardware constraints (e.g., NVIDIA RTX 4090 with 24GB memory). Furthermore, the execution time of our method is comparable to other baselines, as shown in Table 13.

Table 13: Computational cost of distilling CIFAR-100 with storage buffer $N = 100$ using a single Nvidia RTX 4090 GPU card.

| Target | DSA | DM | IDM | DATM | KRR-ST | Ours |
| --- | --- | --- | --- | --- | --- | --- |
| GPU memory (MB) | 3561 | 2571 | 2407 | 4351 | 4483 | 6917 |
| Time (mins) | 81 | 78 | 313 | 121 | 189 | 205 |

| Variable | Meaning |
|---|---|
| $\theta$ | the parameters $(\omega, W)$ of model $\hat{g}_\theta = h_W \circ f_\omega$ |
| $\tau_a$ | the a-th predetermined augmentation |
| $a$ | the index of augmentation |
| $A$ | the number of augmentations |
| $B^x$ | $[b_1^x, \ldots, b_U^x]^\top \in \mathbb{R}^{U \times d_x^b}$ image bases |
| $B^y$ | $[b_1^y, \ldots, b_V^y]^\top \in \mathbb{R}^{V \times d_y}$ representation bases |
| $c$ | number of channels |
| $C^x$ | $\mathbb{R}^{m \times U}$ coefficient for generating distilled images |
| $C^y$ | $\mathbb{R}^{m \times V}$ coefficient for generating distilled representation |
| $C_a^y$ | $\mathbb{R}^{m \times V}$ for $a = 1, \ldots, A$ corresponding to the target representation of $\tau_a(X_s)$ |
| $d_x$ | $c \times h \times w$ the dimension of the sample $x_i$ |
| $d_x^b$ | $c \times \frac{h}{s} \times \frac{w}{s}$ the dimension of image basis |
| $d_y$ | the dimension of representation $y_i$ |
| $f_\omega$ | feature extractor with parameter $\omega$ |
| $g_\phi$ | teacher model |
| $\hat{g}_\theta$ | the model used to mimic the teacher model (usually smaller than $g$) |
| $\hat{g}_{\theta^*}$ | the model used to mimic the teacher model with optimized parameters |
| $h$ | height of image |
| $h_W$ | linear head with parameter $W$ |
| $K_{X_s, X_s}$ | Gram matrix of distilled samples $X_s$ |
| $l$ | the index of model pool |
| $L$ | the size of model pool |
| $\mathcal{M}$ | $\{(\hat{g}_{\theta_1}, z_1), \ldots, (\hat{g}_{\theta_L}, z_L)\}$ model pool |
| $m$ | the number of diltilled data |
| $n$ | the number of real data |
| $N$ | total storage budget |
| $Q_a$ | approximation network corresponding to a-th augmentation |
| $s$ | the scaling factor of the image bases |
| $U$ | the number of image bases |
| $V$ | the number of representation bases |
| $w$ | width of image |
| $x_i$ | real image $i$ |
| $\tilde{x}_i$ | distilled image i |
| $X_s$ | $[\tilde{x}_1, \ldots, \tilde{x}_m]^\top \in \mathbb{R}^{m \times d_x}$ the set of distilled images |
| $X_t$ | $[x_1, \ldots, x_n]^\top \in \mathbb{R}^{n \times d_x}$ the set of real images |
| $X_t'$ | random sample images from $X_t$ |
| $\tilde{X}_s$ | augmented distilled images |
| $\tilde{y}_i$ | target representation for distilled image $\tilde{x}_i$ |
| $Y_s$ | $[\tilde{y}_1, \ldots, \tilde{y}_m]^\top \in \mathbb{R}^{m \times d_y}$ the set of target representations |
| $\tilde{Y}_s$ | the target representation for $\tilde{X}_s$ |
| $\tilde{Y}_s^Q$ | the target representation which approximates by network $\{Q_a\}_{a=1}^A$ for $\tilde{X}_s$ |
| $z_l$ | the trained steps of model $\hat{g}_\theta$ |
| $Z$ | the maximum steps to update the feature extractor |

Table 14: The List of Mathematical Notations

