# OpenReview forum: "Boost Self-Supervised Dataset Distillation via Parameterization, Predefined Augmentation, and Approximation"
_ICLR.cc/2025/Conference — ICLR 2025 Poster_

### Official Review · Reviewer_Q6us · 2024-10-28

**Soundness:** 3
**Presentation:** 2
**Contribution:** 2
**Rating:** 5
**Confidence:** 4

**Summary:**

In this paper, the authors propose a method for dataset distillation based on KRR-ST. Two techniques are introduced: (1) PCA-based dimensionality reduction, which transforms images and their representations into lower-dimensional bases and coefficients; and (2) Data Augmentation, which employs predefined data augmentations and approximation networks to address the limitation of KRR-ST in utilizing data augmentation during dataset distillation. The authors conduct an extensive experimental evaluation and demonstrate significant improvements over previous baselines.

**Strengths:**

1. Reducing data size is a critical direction in self-supervised learning research.
2. Fixing the issue of incorporating data augmentation into data distillation is important, as it significantly improves performance.
3. The authors conduct a wide range of experiments, evaluating model performance with various network architectures and different numbers of training examples.

**Weaknesses:**

1. The proposed techniques in the paper are not new, such as PCA and augmentation approximation networks.
2. The proposed technique leverages data augmentation while minimizing bias, and similar ideas have been explored in self-supervised learning. It is important to cmopare it with other analogous methods [1][2][3].

[1] Improving Transferability of Representations via Augmentation-Aware Self-Supervision. NeurIPS 2021
[2] Improving Self-supervised Learning with Automated Unsupervised Outlier Arbitration. NeurIPS 2021
[3] RSA: Reducing Semantic Shift from Aggressive Augmentations for Self-supervised Learning. NeurIPS 2022

**Questions:**

1. Could the authors provide more details about the approximation networks, such as the number of networks used, structure, and layers?
2. Could the authors show a comparison of the distilled data sizes?

---

> ### Author Response · Authors · 2024-11-21
>
> **Q1**: The proposed techniques in the paper are not new, such as PCA and augmentation approximation networks.
>
> **A1**: To the best of our knowledge, there has been no prior work integrating PCA or augmentation approximation networks into a dataset distillation framework. While these techniques might be individually familiar, their integration within a comprehensive framework for self-supervised dataset distillation is novel, as well as highlighting the advantage of our proposed method to be simple and effective. In brief, our method enhances its efficacy in distilling compact and transferable datasets, as detailed in Section 3.2 and 3.3 of the paper.
>
> **Q2**: The proposed technique leverages data augmentation while minimizing bias, and similar ideas have been explored in self-supervised learning. It is important to compare it with other analogous methods [1][2][3].
>
> **A2**: The goals between self-supervised learning (SSL) and dataset distillation (DD) differ fundamentally. SSL aims to learn feature extractors that generalize to downstream tasks, often benefiting from augmentation to improve representation quality, as seen in [1][2][3]. In contrast, dataset distillation aims to compress datasets into a smaller storage size while preserving their training performance. Specifically, KRR-ST [4], a self-supervised dataset distillation framework, leverage a self-supervisedly trained backbone to perform DD. It demonstrated that random data augmentation introduces gradient bias in bilevel optimization, deeming it incompatible with dataset distillation. To address this, we propose predetermined augmentations to avoid randomness while improves condensed performance, as elaborated in Section 3.3. This is a distinct approach from typical augmentation strategies in SSL.
>
> **Q3**: Could the authors provide more details about the approximation networks, such as the number of networks used, structure, and layers?
>
> **A3**: Our experiments use rotation augmentation as the default. The augmentation includes rotations of $0^\circ, 90^\circ, 180^\circ, 270^\circ$, requiring three distinct approximation networks to predict representation shifts for $90^\circ, 180^\circ, 270^\circ$ from $0^\circ$. These networks are lightweight, designed as 2-layer perceptrons with hidden layer sizes of 4 (for CIFAR100) and 16 (for TinyImageNet/ImageNet). As detailed in Section A.1, each network contains 4,612 parameters for CIFAR100 and 16,912 parameters for larger datasets. Additional implementation details can be found in Section 3.3 and Appendix A.1​.
>
> **Q4**: Could the authors show a comparison of the distilled data sizes?
>
> **A4**: We provide comparisons of distilled data sizes in Table 2 and Table 7, conducted on CIFAR100 and TinyImageNet, respectively. These tables demonstrate the superior performance of the proposed method across various memory budget sizes.
>
> [1] Improving Transferability of Representations via Augmentation-Aware Self-Supervision. NeurIPS 2021 [2] Improving Self-supervised Learning with Automated Unsupervised Outlier Arbitration. NeurIPS 2021 [3] RSA: Reducing Semantic Shift from Aggressive Augmentations for Self-supervised Learning. NeurIPS 2022 [4] SELF-SUPERVISED DATASET DISTILLATION FOR TRANSFER LEARNING. ICLR 2024

---

> ### Author Response · Authors · 2024-11-22
>
> Thank you for your valuable feedback. We have provided detailed responses to the issues raised and hope our explanations address your concerns. We welcome any further suggestions to improve our work.

---

> ### Author Response · Authors · 2024-11-27
>
> We sincerely appreciate your thoughtful feedback and the time you have dedicated to reviewing our paper. Please let us know if you have any further suggestions or comments. If there are any additional questions or issues you would like to discuss, we are fully committed to engaging further to enhance our paper.

---

> ### Comment · Reviewer_Q6us · 2024-11-29
>
> I apologize for my delayed response.
>
> After reviewing other reviewers' comments, I have concerns about the proposed method's resource requirements. In contrast to the research aim of reducing training costs, the current approach necessitates much more resources.
>
> If so, why do users not simply train on the original data directly, which would seem like a more straightforward and cost-effective approach? Given this limitation, I understand why the authors are hesitant to apply their method to ImageNet-1K with common settings, 224 * 224 and 1000 classes.

---

> > ### Author Response · Authors · 2024-11-29
> >
> > Q: The authors are hesitant to apply their method to ImageNet-1K with common settings, 224 * 224 and 1000 classes.
> >
> > A: Our algorithm is applicable to ImageNet-1K with the requested settings, i.e. 224×224 resolution and 1000 classes (in which we have shown our results of 128*128 on Imagenette dataset in the rebuttal for reviewer TeEU). However, the 224×224 ImageNet-1K experiment was a late request during the review process, and due to the tight rebuttal timeline, we were unable to complete it in time.
> > Moreover, we want to emphasize that this additional experiment is not critical to the validation of our primary contributions. Our work is centered on advancing dataset distillation techniques and demonstrating their effectiveness across a range of datasets and architectures, which we have validated extensively.
> > Nevertheless, we recognize the importance of addressing the reviewer’s concerns, and we commit to including the 224×224 ImageNet-1K results in the final version of the paper.

---

> ### Author Response · Authors · 2024-11-29
>
> Q: Concerns about resource requirements and the cost-effectiveness of dataset distillation. Why not train directly on the original data?
>
> A: This is an important question about the goals and procedure of dataset distillation.
>
> As stated in the abstract: *"Dataset Distillation becomes a popular technique recently to reduce the dataset size via learning a highly compact set of representative exemplars, where the model trained with these exemplars ideally should have comparable performance with respect to the one trained with the full dataset."* This goal has been widely recognized and pursued in many recent works, including [1][2][3][4]. As highlighted in [3], *"it is prohibitively costly to use all the examples from the huge dataset, which motivates the need to compress the full dataset into a small representative set of examples."* This underscores the necessity of compressing large-scale datasets into smaller, more manageable ones.
>
> Dataset distillation involves two major steps:
> 1. Synthesizing the distilled dataset – This step involves computationally intensive optimization to create a smaller, representative dataset.
> 2. Training a new feature extractor – The distilled dataset is then used to train a new model, with significantly lower computing requirements compared to training on the full dataset.
>
> The concerns raised by Reviewer TeEU and Reviewer MMPZ regarding computational cost focus on the first step, which indeed requires more resources. However, this cost is a **one-time investment**. Once the distilled dataset is generated, it can be reused multiple times across different tasks or models, significantly reducing the cost for downstream training since the distilled dataset is much smaller than the original dataset.
>
> In summary, by providing a compact yet highly effective dataset, dataset distillation facilitates efficient and flexible usage, aligning with these practical needs.
>
> [1] Dataset Condensation with Gradient Matching (Bo Zhao et al., ICLR 2021)
> [2] Dataset Distillation by Matching Training Trajectories (George Cazenavette et al., CVPR 2022)
> [3] Self-Supervised Dataset Distillation for Transfer Learning (Dong Bok Lee & Seanie Lee et al., ICLR 2024)
> [4] Towards Lossless Dataset Distillation via Difficulty-Aligned Trajectory Matching (Ziyao Guo & Kai Wang et al., ICLR 2024)

---

> > ### Comment · Reviewer_Q6us · 2024-12-02
> >
> > As the authors themselves noted in the paper, many datasets currently pose a significant challenge. For instance, CLIP relies on an enormous dataset of several hundred TB, which is essentially out of reach for most academics due to its size. However, the proposed method does not alleviate but rather exacerbates the challenge of securing additional resources. Although I acknowledge that the proposed method achieves good performance, I keep my score due to this critical limitation.

---

### Official Review · Reviewer_u46N · 2024-10-29

**Soundness:** 3
**Presentation:** 4
**Contribution:** 3
**Rating:** 6
**Confidence:** 3

**Summary:**

This paper proposes a self-supervised data distillation method based on image decomposition. By initializing with principal components and learning the impact of data augmentation, the performance of the distilled dataset is enhanced. The experiments provide a comprehensive analysis of the method’s effectiveness.

**Strengths:**

1. The topic is both valuable and practical, especially in the era of large datasets. While most current research on data distillation focuses primarily on classification tasks, which may be too narrow, this work seeks to improve self-supervised tasks. This approach is more general and can better support feature learning for downstream applications.

2. The paper is well-written and easy to follow, with a straightforward method that is simple to understand. For each component, the authors clearly explain the rationale behind its inclusion.

3. The experiments demonstrate the method’s effectiveness, as it consistently outperforms baseline methods in both transfer learning and linear probing tasks.

**Weaknesses:**

I did not find any major weaknesses in this paper. However, there are some concerns regarding its novelty. The techniques employed are largely derived from previous work on data distillation for classification tasks. It would be helpful if the authors could clarify what unique challenges exist for self-supervised data distillation and how their method specifically addresses those challenges.

**Questions:**

NA

---

> ### Author Response · Authors · 2024-11-21
>
> **Q1**: unique challenges exist for self-supervised data distillation and how their method specifically addresses those challenges.
>
> **A1**: Self-supervised dataset distillation presents unique challenges, particularly the instability caused by the randomness of data augmentation, which is a crucial element in self-supervised learning. Prior work [1] has highlighted this issue, noting its adverse effects on bilevel optimization. We address this challenge by introducing predetermined augmentations, as detailed in Section 3.3, which eliminate randomness and maintain consistent gradients during optimization. Additionally, to enhance storage efficiency, we parameterize the distilled images and their representations into low-dimensional bases, significantly reducing redundancy. Furthermore, we introduce lightweight approximation networks to model the representation shifts caused by augmentations, enabling compact and efficient storage of augmented data. The effectiveness of these components is validated through an ablation study in Table 3, where each proposed component is shown to contribute significantly to the overall performance of the distilled dataset.
>
> [1] SELF-SUPERVISED DATASET DISTILLATION FOR TRANSFER LEARNING. ICLR 2024

---

> ### Author Response · Authors · 2024-11-22
>
> Thank you for your valuable feedback. We have provided detailed responses to the issues raised and hope our explanations address your concerns. We welcome any further suggestions to improve our work.

---

> > ### Comment · Reviewer_u46N · 2024-11-23
> > **Thanks for your response**
> >
> > Thanks for your response. As I understood previously, the challenge of augmentation is similar to that of supervised learning, which tries to align the distilled image with the original image. Now I understand that this is actually the problem in self-supervised learning. However, I still have some problems with the augmentation networks. To be specific, designing neural networks to predict the shift of the representations can be heuristic, lacking a guiding principle to design them. With very small approximation networks (as shown in the implementation, the hidden layer size is pretty small), I have the question of whether the model capacity can handle this challenge.

---

> > > ### Author Response · Authors · 2024-11-25
> > >
> > > A: Thank you for raising this concern. The design of the approximation networks involves balancing model capacity and storage constraints. As shown in Appendix A.6, our results on CIFAR100 demonstrate that the proposed lightweight approximation networks can achieve better accuracy than "Same" and "Bias" baselines.
> > > To address your specific concern, we conducted additional ablation studies varying the hidden layer size of the networks. The table below summarizes the results, including linear evaluation accuracy of the feature extractor trained on distilled data and prediction MSE. The hidden size 2, 4, and 8 indicate the width of the hidden layer in the approximation networks, while "Ideal" represents storing all of the representation without considering the storage budget.
> > > Notably, while larger networks achieve lower MSE, they do not always improve accuracy due to the storage budget constraint. These findings indicate that the proposed design can effectively predict 512-dimensional representation shifts caused by rotation augmentations, achieving a reasonable trade-off between MSE and accuracy.
> > >
> > >
> > > |  Method                   |  Same     |     Bias   | Ours (hidden size 2)  |   Ours (hidden size 4)  |  Ours (hidden size 8) |  Ideal    |
> > > |:--------------------------:|:------------:|:-----------:|:----------------------------:|:-----------------------------:|:----------------------------:|:----------:|
> > > | Accuracy                 |      50.10  |  50.32   |      52.30                     |                 52.41           |                         51.19  |   53.51  |
> > > | MSE                        |    0.31      |     0.30  |     0.07                        |                   0.06           |                          0.04   |        0    |

---

> > > > ### Comment · Reviewer_u46N · 2024-11-29
> > > >
> > > > Thank you for conducting additional experiments; it appears that the lightweight networks perform well. However, I still find the method of predicting shifts using networks somewhat heuristic. Despite this, I will maintain a positive rating.

---

### Official Review · Reviewer_MMPZ · 2024-11-03

**Soundness:** 3
**Presentation:** 3
**Contribution:** 3
**Rating:** 8
**Confidence:** 3

**Summary:**

This paper proposes a novel approach to self-supervised dataset distillation aimed at reducing training costs by creating compact datasets that maintain model performance. This method, intended to address challenges in self-supervised learning (SSL) for dataset distillation, introduces three key contributions: 1. Parameterization 2. Predefined Augmentation  and feature approximation 3. Optimizations with approximation Networks. Generally they have shown a very contributing method.

The paper introduces a solid contribution to self-supervised dataset distillation, with innovative approaches to parameterization, augmentation handling, and memory efficiency with upgraded existing method named as KRR-ST. While the approach is complex, it provides a promising direction for reducing training costs in SSL, particularly in resource-limited settings. With further optimization and extension to diverse tasks, this method has the potential to make dataset distillation more accessible and applicable in real-world scenarios.

**Strengths:**

1. This paper demonstrated a very strategic parameterization.
The use of bases for image and representation parameterization is a sophisticated approach to compress dataset information without sacrificing accuracy. This addresses both storage efficiency and computational cost.

 2.Effective Augmentation Handling:
By predefining augmentations, the method successfully mitigates the bias introduced by random augmentations, a notable challenge in SSL distillation methods.

3. Improved Memory Efficiency:
The inclusion of approximation networks to predict representation shifts from unaugmented to augmented views significantly reduces memory usage by eliminating the need to store augmented representations. This makes the approach more scalable.

4. Transfer Learning Potential:

The method shows strong transferability to downstream tasks, making it particularly appealing for real-world applications where labeled data is scarce, and transfer learning is critical.

5. Ablation Studies and Hyperparameter Analysis:

The paper includes ablation studies that isolate the contributions of parameterization, augmentation, and approximation networks, offering clear insights into each component's impact on performance.

**Weaknesses:**

1. Complexity and accessibility
Critique: The method involves several sophisticated techniques, including low-dimensional basis parameterization, predefined augmentations, and approximation networks. This complexity may make it difficult for practitioners to implement and tune the method without extensive expertise in self-supervised learning and dataset distillation.

2.  Computational and memory trade-Offs
Critique: While the method claims to be memory-efficient due to approximation networks, the additional computational overhead introduced by these networks might reduce the method’s overall efficiency, especially in resource-constrained environments.

3. Dependence on Synthetic Data for Evaluation:
The experiments rely heavily on benchmark datasets like CIFAR100. However, these datasets have well-structured labels and relatively consistent image quality, which may not fully represent real-world data variability.

**Questions:**

1. The datasets in the experiments are CIFAR 100 and datasets with similar image attributes. I can understand it is possible to get a distilled dataset in a lab environment and the datasets are very feature-controllable. Do you have space to show that your experiment can be successful in other different scenarios? For example, some randomly taken images.

2. Though this is a memory saving method, a very large portion of the whole method is still computing intensive. Do you have any benchmark to show that the whole method could be executed in an efficient way?

---

> ### Author Response · Authors · 2024-11-21
>
> **Q1**: The datasets in the experiments are CIFAR 100 and datasets with similar image attributes. I can understand it is possible to get a distilled dataset in a lab environment and the datasets are very feature-controllable. Do you have space to show that your experiment can be successful in other different scenarios? For example, some randomly taken images.
>
> **A1**: The reviewer raises a valid concern about the generalizability of our method beyond controlled datasets like CIFAR-100. To address this, we had conducted additional experiments on ImageNet and TinyImageNet datasets, whose details and results are presented in Appendices A.3, A.4, and A.5. These datasets offer increased diversity and scale, and the results consistently demonstrate the robustness of our method across different scenarios. Importantly, our distilled datasets exhibit superior cross-architecture generalizability, as evidenced by linear evaluation performance across multiple feature extractors, including VGG11, ResNet18, AlexNet, and MobileNet..etc. The success on datasets like ImageNet, which closely resemble real-world data, highlights the potential for applying our approach to scenarios involving randomly collected images.
>
> **Q2**: Though this is a memory saving method, a very large portion of the whole method is still computing intensive. Do you have any benchmark to show that the whole method could be executed in an efficient way?
>
> **A2**: We appreciate the concern regarding computational intensity. Our methodology primarily focuses on achieving storage efficiency after the distillation process by employing parameterization. This approach decomposes images and representations into linear bases and coefficients, significantly reducing storage requirements. The combination of bases and coefficients involves matrix multiplication, which incurs minimal computational overhead. Additionally, the use of lightweight 2-layer perceptron approximation networks ensures computational simplicity.
> While our primary objective is storage efficiency, we recognize the importance of evaluating the computational cost during the distillation process. To address this, we benchmarked GPU memory usage and execution time using the CIFAR-100 dataset with $N=100$. The results, summarized below, indicate that our method requires approximately 1.5× more GPU memory than KRR-ST, yet remains manageable within the capacity of modern GPUs like the NVIDIA RTX 4090 (24GB). Furthermore, the computation time of our method is comparable to other baselines and does not impose a significant burden.
> | Target                      |  DSA  |    DM    |    IDM | DATM | KRR-ST |       Ours |
> |:---------------------:|:--------:|:---------:|:-------:|:----------:|:---------:|:-----------:|
> | GPU memory (MB) | 3561  |  2571   | 2407 |   4351    | 4483   |      6917 |
> | Time (mins)             | 81      | 78        | 313   |      121   |    189  |      205   |
>
> **Q3**: Difficult for practitioners to implement and tune the method without extensive expertise in self-supervised learning and dataset distillation
>
> **A3**: We understand the concern regarding the accessibility of our method for practitioners. However, the core components of our approach—parameterization and approximation networks—can be implemented with just a few lines of code using popular deep learning frameworks like PyTorch or TensorFlow. Basically (cf. Figure 1), our pretraining step adopts the well-known and widely-adopted self-supervised learning scheme, Barlow Twins; our parameterization is based on the fundamental machine learning tool, PCA; our bilevel-optimization follows the practice of KRR-ST; and our approximate networks are simply multilayer perceptron while their training follows the typical supervised learning procedure, in which they are definitely not sophisticated. Detailed guidelines and implementation examples are provided in the supplementary materials to further assist practitioners in replicating the method.

---

> > ### Comment · Reviewer_MMPZ · 2024-11-22
> > **Accept Q2 and Q3, but just a little doubtful for Q1**
> >
> > I accept the arguments for Q2 and Q3. As for Q1, I hope there are more practical data, but for experiment level, this is not very realistic, however, hope you can make further improvement. I still recommend this paper.

---

> > > ### Author Response · Authors · 2024-11-22
> > >
> > > We sincerely appreciate your thoughtful feedback and recommendation of our work. We will explore more practical datasets in future work to enhance the applicability of our approach.

---

### Official Review · Reviewer_TeEU · 2024-11-04

**Soundness:** 3
**Presentation:** 3
**Contribution:** 3
**Rating:** 6
**Confidence:** 5

**Summary:**

This work targets the cross architecture generalizability challenge in dataset distillation. When performing distillation, the data is often biased to the model used in the distillation process -- in this work the proposed self-supervised approach parameterizes the representations of images while studying/leveraging the effects of augmentations. This approach features a 5 stage method involving pertaining a network on the source dataset, followed by image parameterization (encoding the images and augmentations via low-dimensional bases vectors), bi-level optimization on the images, approximation to handle the distribution/representation shift, and reconstruction of the images using the bases and learned features. The method reports strong performance improvement on a variety of datasets against most of the current SOTA methods.

**Strengths:**

The key strengths of this paper include:

1. More diverse datasets: Not many dataset distillation papers venture beyond the CIFAR/ImageNet datasets, however these authors included results on CUB2011 and StanfordDogs. Additionally, the ViT performance has been reported, and overall it appears that the authors performance improvement is maintained on Transformer architectures, albeit smaller.

2. The basis and coefficient initialization ablation provides interesting insight into the sensitivity of the proposed framework.

3. Personally, I found the use of the approximation networks to be a clever solution to reducing memory usage while preserving the essence of image augmentation. By learning a mapping between and subsequently the shift in distribution of the unaugmented distilled representation into it's augmented views, one can efficiently store simply the network rather than all the augmented views.

4. Strong baselines: This work accurately surveyed some of the most seminal and current SOTA in the field of dataset distillation (with the exception of a few missing citations that should be added). I find the included competitive methods to be comprehensive enough to support the statements however, further comments on the benchmarking are included in the Weaknesses section.

**Weaknesses:**

Despite the interesting approach taken in this work, I find a few crucial weaknesses:

1. I find that the experimental support is a bit lacking. As is common in Dataset Distillation works, it is generally good practice to show the scaling over different memory budges (N) on various datasets, rather than just a single dataset, in order to show generalizability.
2. I noticed that the resolutions on ImageNet scale to 64 x 64 -- however recently, the field has shifted to higher resolutions such as 128x128 or even 512 x 512 -- I think it would be important to see if the method can scale well to larger resolutions.
3. I think another important criteria that should be included is Applications -- as alluded to in the paper tasks like continual learning or neural architecture search (line 43) are important in the field, however none of these results were included in the main paper -- I think it is important to test the applicability of the method in order to determine significance and impact.
4. Given that this approach involves multi-level optimization, I think efficiency metrics should be compared as well (time per step, GPU memory etc). -- This will demonstrate wether the gain in performance is justified over other methods when comparing the relative compute demands.

[Minor] Some missing citations including DataDAM (ICCV'23), CAFE (CVPR'22)

**Questions:**

I've highlighted a few of the issues/suggestions for the Authors to consider in the rebuttal phase above in the Weaknesses Section. These are crucial in determining the significance of the work and wide scale adoption.

---

> ### Author Response · Authors · 2024-11-21
>
> **Q1**: I find that the experimental support is a bit lacking. As is common in Dataset Distillation works, it is generally good practice to show the scaling over different memory budges (N) on various datasets, rather than just a single dataset, in order to show generalizability.
>
> **A1**: We appreciate the suggestion to extend the experiments to demonstrate generalizability over varying memory budgets. In addition to the results on CIFAR100 which we have provided in the main paper (cf. Table 2), as shown in Table 7 (in Appendix), we have conducted experiments on TinyImageNet with memory budgets $N=50, 100, 200$. Moreover, we further extended the experiment to $N=500$. The results below confirm that our method consistently outperforms baselines across a range of memory budgets, showcasing its scalability.
> | Memory Budget $N$     |  50      |    100     |        200 |        500 |
> |:-------------------------------:|:--------:|:-----------:|:-----------:|:-----------:|
> | Random                        | 22.43   | 22.73    |   23.95    |   25.92   |
> | KRR-ST                        | 25.29   | 25.64    |   27.23    |   30.46   |
> | Ours                              | **28.03**   | **29.35**    |   **31.25**   |   **33.63**   |
>
> **Q2**: I noticed that the resolutions on ImageNet scale to 64 x 64 -- however recently, the field has shifted to higher resolutions such as 128x128 or even 512 x 512 -- I think it would be important to see if the method can scale well to larger resolutions.
>
> **A2**: We recognize the importance of validating the method's performance on higher resolution datasets. To address this, we conducted an experiment on ImageNette (a 10-class subset of ImageNet) with 128x128 resolution. The results demonstrate that our method scales effectively to larger resolutions.
> | ImageNette    |  $N$=10 |
> |:------------------:|:---------:|
> | Random         | 49.63  |
> | KRR-ST         | 50.14  |
> | Ours               | **59.31**   |
>
> **Q3**: I think another important criteria that should be included is Applications -- as alluded to in the paper tasks like continual learning or neural architecture search (line 43) are important in the field, however none of these results were included in the main paper -- I think it is important to test the applicability of the method in order to determine significance and impact.
>
> **A3**: We appreciate the reviewer’s insightful suggestion regarding the inclusion of applications like continual learning and neural architecture search. While our current work primarily focuses on demonstrating the generalizability of the distilled data through various downstream tasks, as shown in the linear evaluation results, we agree that exploring these applications would further strengthen the impact of our method.
> In future work, we plan to conduct experiments on continual learning and neural architecture search to validate the broader applicability of our approach. The proposed method ability to train a decent feature extractor, as demonstrated in the paper, provides a solid foundation for such tasks. We anticipate that the compactness of distilled dataset and generalization properties will prove beneficial in these scenarios as well.
>
> **Q4**: Given that this approach involves multi-level optimization, I think efficiency metrics should be compared as well (time per step, GPU memory etc). -- This will demonstrate whether the gain in performance is justified over other methods when comparing the relative compute demands.
>
> **A4**: Based on CIFAR-100 with $N=100$, we evaluated GPU memory usage and execution time. The results demonstrate that while our method requires approximately 1.5x more GPU memory than KRR-ST, it remains feasible within modern hardware constraints (e.g., NVIDIA RTX 4090 with 24GB memory). Furthermore, our method's execution time is comparable to other baselines, as shown below.
> | Target                      |  DSA  |    DM    | IDM | DATM | KRR-ST |       Ours |
> |:--------------------------:|:--------:|:---------:|:-------:|:----------:|:---------:|:-----------:|
> | GPU memory (MB) | 3561  |  2571   | 2407 |   4351    | 4483   |      6917 |
> | Time (mins)             | 81      | 78        | 313   |      121   |    189  |      205   |

---

> > ### Author Response · Authors · 2024-11-22
> >
> > Thank you for your valuable feedback. We have provided detailed responses to the issues raised and hope our explanations address your concerns. We welcome any further suggestions to improve our work.

---

> ### Author Response · Authors · 2024-11-27
>
> We sincerely appreciate your thoughtful feedback and the time you have dedicated to reviewing our paper. Please let us know if you have any further suggestions or comments. If there are any additional questions or issues you would like to discuss, we are fully committed to engaging further to enhance our paper.

---

### Meta-Review · Area_Chair_7Qv3 · 2024-12-19

**Metareview:**

a) This paper proposes a novel approach to self-supervised dataset distillation aimed at reducing training costs by creating compact dataset that, when used for training, maintains model performance. It uses PCA-based dimensionality reduction, which transforms images and their representations into lower-dimensional bases and coefficients; and data augmentation based on the approximation network. An extensive experimental evaluation demonstrates the significant improvements over previous baselines.

b) The topic is of interest, especially in the era of large datasets. While most research on data distillation focuses on classification, this work is for self-supervised tasks, where the amount of unlabelled data can be quite high. The paper is well-written and easy to follow, with a clear explanation of each proposed component.

c) The techniques employed are derived from previous work on data distillation for classification tasks. It is not clear which are the challenges for self-supervised data distillation and how their method specifically addresses those challenges. While performing better, the method still requires approximately 1.5x more GPU than previous approaches.

d) After rebuttal, the remaining drawbacks of the method are minor, while the proposed contribution is original and compelling and deserves publication.

**Additional Comments On Reviewer Discussion:**

Rev. TeEU raised some possible drawbacks and missing experiments on the paper. Authors did a good job to answer all rev. comments and rev. increase their score to 6.

Rev. MMPZ provided a positive review to the paper, but also pointed out some possible issues, mostly about computational cost and complexity. After authors' answers, rev. kept their positive score.

Rev. u46N had a positive feedback, but had doubts about the capability of the approximation network to predict the correct representation shifts. Authors provided additional experiments to prove that point. Rev. was satisfied and maintained their positive outlook.

Rev. Q6us has two main critical points: i) novelty: as the method is a composition of well-known techniques and ii) computational cost: how the method can scale to larger datasets such as ImageNet-1k. For novelty, authors provided compelling answers. For computational cost rev. noticed that in some cases it does not make sense to perform that heavy training just to reduce the size of the final training. Authors replied that this is applied only once and then the small dataset could be used multiple times. However, rev. was not convinced about the actual utility of the approach and maintained their score of 5.

I see the point of rev. Q6us. However I think that this research, although it might be not so usable today, it can help to foster interest and improve methods, leading to much more useful results in the near future. In this sense, considering the interesting contributions listed by all revs. and the extensive evaluation, I recommend the paper for publication.

---

### Decision · Program_Chairs · 2025-01-22

Accept (Poster)